# Dilp-2–mediated PI3-kinase activation coordinates reactivation of quiescent neuroblasts with growth of their glial stem cell niche

Xin Yuan[1], Conor W. Sipe[1,2], Miyuki Suzawa[3], Michelle L. Bland[3], Sarah E. Siegrist[1]*

1 Department of Biology, University of Virginia, Charlottesville, Virginia, United States of America,
2 Department of Biology, Shepherd University, Shepherdstown, West Virginia, United States of America,
3 Department of Pharmacology, University of Virginia, Charlottesville, Virginia, United States of America

* ses4gr@virginia.edu

**Data Availability Statement:** All relevant data are within the paper and its Supporting Information files.

## Abstract

Dietary nutrients provide macromolecules necessary for organism growth and development. In response to animal feeding, evolutionarily conserved growth signaling pathways are activated, leading to increased rates of cell proliferation and tissue growth. It remains unclear how different cell types within developing tissues coordinate growth in response to dietary nutrients and whether coordinated growth of different cell types is necessary for proper tissue function. Using the early *Drosophila* larval brain, we asked whether nutrient-dependent growth of neural stem cells (neuroblasts), glia, and trachea is coordinated and whether coordinated growth among these major brain cell types is required for neural development. It is known that in response to dietary nutrients and PI3-kinase activation, brain and ventral nerve cord neuroblasts reactivate from quiescence and ventral nerve cord glia expand their membranes. Here, we assay growth in a cell-type specific manner at short time intervals in the brain and determine that growth is coordinated among different cell types and that coordinated growth is mediated in part through activation of PI3-kinase signaling. Of the 7 *Drosophila* insulin-like peptides (Dilps), we find that Dilp-2 is required for PI3-kinase activation and growth coordination between neuroblasts and glia in the brain. Dilp-2 induces brain cortex glia to initiate membrane growth and make first contact with quiescent neuroblasts. Once reactivated, neuroblasts promote cortex glia growth to ultimately form a selective membrane barrier. Our results highlight the importance of bidirectional growth signaling between neural stem cells and surrounding cell types in the brain in response to nutrition and demonstrate how coordinated growth among different cell types drives tissue morphogenesis and function.

**Funding:** This work was funded by National Institutes of Health: F32-NS096919 to CWS, R01-DK099601 to MLB, and R00-HD067293 and R01-GM120421 to SES. The funders had no role in study design, data collection and analysis, decisions to publish, or preparation of the manuscript.

**Competing interests:** The authors have declared that no competing interests exist.

**Abbreviations:** Akt/PKB, Protein Kinase B; ALH, after larval hatching; BCA, bicinchoninic assay; BBB, blood–brain barrier; btl, *breathless*; CNS, central nervous system; CSF, cerebral spinal fluid; Dilp, *Drosophila* insulin-like peptide; Dpn, Deadpan; dp60, *Drosophila protein 60*; DRN, Dilp-recruiting neuron; dupRNAi, *double-parked* RNA interference; EdU, 5-ethynyl-2′-deoxyuridine; FDS, fat-body–derived signal; Foxo, forkhead box, subgroup O; Gapdh1, Glyceraldehyde-3-phosphate dehydrogenase 1; GFP, green fluorescent protein; IGF, insulin-like growth factor; InR, insulin-like tyrosine kinase receptor; IPC, insulin-producing cell; MB, mushroom body; mCD8, membrane-targeted CD8 antigen; pcna, proliferating cell nuclear antigen; PDGFR, Platelet-derived growth factor receptor; PEM, Pipes, EGTA, magnesium chloride; PIP$_3$, phosphatidylinositol (3,4,5)-triphosphate; PI3-kinase, phosphoinositide 3-kinase; Pten, phosphatase and tensin homolog; PVR, PDGF- and VEGF-receptor related; raptorRNAi, *raptor* RNA interference; repo, *reversed polarity*; RFP, red fluorescent protein; Scrib, Scribble; SPG, subperineurial glia; TOR, target of rapamycin; UAS, upstream activating sequence; VNC, ventral nerve cord; wor, *worniu*.

## Introduction

Organs must be appropriately sized and patterned to function properly and meet physiological needs of adult animals. To achieve proper organ size and function, a multitude of different cell types are produced over time and in space. Yet, it remains unclear how different cell types with different molecular programs integrate their own growth with that of their nearby neighbors, some of which have different developmental origins. This is particularly true for stem cells, many of which reside within specialized microenvironments composed of many different cell types. Hormones and other secreted growth factors and cytokines have clear and important roles in growth regulation from the single-cell to tissue level and beyond [1–6]. What is less understood is whether local growth decisions are coordinated in response to nutrient availability, what regulatory networks coordinate local growth, and whether coordinated growth is required for tissue function. Dietary nutrients also play important roles in growth regulation because they provide the building blocks for biosynthesis of macromolecules (lipids, proteins, and nucleic acids) and serve as cofactors used in metabolic reactions.

Here, we investigate how growth occurs in the *Drosophila* brain in response to dietary nutrients during development. In *Drosophila*, nutrient-sensing pathways, including insulin/phosphoinositide 3-kinase (PI3-kinase) and target of rapamycin (TOR)-kinase, are evolutionarily conserved, and genetic tools for cell-type–specific manipulation of gene function are available [7–9]. The *Drosophila* brain is well-suited for investigating how growth is regulated and coordinated among different cell types in response to dietary nutrients because it contains a multitude of different cell types, including a heterogeneous population of neural stem cells and progenitors, a diverse array of specialized glia, different neuron subtypes including neurosecretory neurons, and a tracheal network of branched epithelial tubules for oxygen exchange. Furthermore, in *Drosophila*, the time when animals receive their first dietary nutrients can be precisely controlled and the growth of molecularly and functionally distinct cell types assayed over time using available markers. Better understanding of how dietary nutrients regulate brain growth and development is fundamentally important given the incidence of human neurodevelopmental disorders and defects associated with poor nutrient quality and/or limited nutrient availability [10–14].

*Drosophila* receive their first dietary nutrients after consuming their first meal as freshly hatched larvae, which occurs about 22 hours (at 25˚C) after fertilization following completion of embryogenesis. Dietary amino acids stimulate TOR-kinase pathway activity in the fat body, a lipid storage organ with endocrine function similar to the mammalian liver [15,16]. In response to dietary nutrients, fat-body–derived signals (FDSs) are synthesized and secreted into the circulating hemolymph, which stimulates tissue growth systemically [16–21]. Subperineurial glia (SPG), a glial subtype that forms part of the blood brain-like barrier (BBB), respond to the FDS by synthesizing and secreting *Drosophila* insulin-like peptide 6 (Dilp-6) locally within the central nervous system (CNS) [21–23]. Local Dilp-6 activates the PI3-kinase pathway in neural stem cells, known as neuroblasts in *Drosophila*, leading to increased growth and their reactivation from developmental quiescence [21,22]. This is the current model for how dietary nutrient conditions regulate neuroblast reactivation from developmental quiescence. What is lacking is a better understanding of how other cell types within the brain grow in response to dietary nutrients, whether growth among these cell types is coordinated in response to dietary nutrients, and whether their growth is required for neuroblast reactivation. Stem cells generally reside within specialized microenvironments, referred to as niches, that support and insulate them from outside factors [24]. These specialized microenvironments integrate local and systemic cues to control stem cell proliferation decisions, cues that report on nutrient status, tissue physiology and function, and perhaps even developmental time.

Previous work has demonstrated that neuroblasts reactivate from developmental quiescence in a nutrient and PI3-kinase dependent manner [17, 21–23]. More recently, development of ventral nerve cord cortex glia, a CNS glia subset, was reported to occur in three steps during larval stages [23]. During the first third of larval development, which corresponds to approximately 32–40 hours ALH (after larval hatching), cortex glia in the ventral nerve cord expand their membranes [23]. Then, after expansion, cortex glia encase neuroblasts to form checkerboard-like chambers [23]. Then, cortex glia undergo extension to accommodate increases in chamber size due to continued NB proliferation [23]. During expansion and before encasement, neuroblasts are reported to lie adjacent to BBB glia, a different CNS glial subset that synthesizes and secretes Dilp6 in a nutrient-dependent manner [23]. It is thought that absence of cortex glia encasement allows neuroblasts to directly receive Dilp6 from BBB glia and reactivate from developmental quiescence [23]. Chamber formation, like neuroblast reactivation, was reported to be nutrient-regulated, and when PI3-kinase levels are reduced, neuroblast encasement is incomplete and neuron apoptosis increased [23]. Moreover, it was reported that activation of PI3-kinase signaling in ventral nerve cord neuroblasts, through expression of constitutively-active Akt, led to cortex glial chamber formation in the absence of dietary nutrients [23].

Here we assay growth dynamics of different cell types within the central brain during the first 24 hours of larval development, a time that most closely corresponds to the expansion phase described above. We precisely correlate neuroblast growth and S-phase entry with changes in glial and tracheal membrane growth and endoreplication. By assaying cell-type specific changes at short, defined intervals during early larval stages, we find that growth of all three cell types occurs continuously and concomitantly, suggesting that common nutrient-sensing pathways regulate and coordinate growth among the different cell types in response to nutrition. We find that *Drosophila* insulin-like peptide 2 (Dilp-2) mediates PI3-kinase activation in neuroblasts and glia and that Dilp-2–mediated PI3-kinase activity is required to coordinate the growth of central brain neuroblasts with the development of their cortex glial niche. Cortex glia, a subset of brain glia, ensheathe neuroblasts and their progeny with a selective membrane that provides barrier function to protect neuroblasts and their progeny from outside factors. Furthermore, we find that Dilp-6 is dispensable for neuroblast reactivation from quiescence, as well as the growth of glia and trachea in the brain, compensated for by high *dilp2* transcript and protein levels in response to feeding. This work highlights the importance of bidirectional growth signaling among different cell types in response to nutrition and sheds light on how dietary nutrients coordinate growth among different cell types within the developing *Drosophila* brain.

## Results

### Neuroblast reactivation from quiescence, glial growth, and tracheal morphogenesis are nutrient regulated

To better understand how cell growth is coordinated with tissue growth in response to dietary nutrients, we assayed proliferation and growth rates of different cell types in the *Drosophila* brain in response to animal feeding. Freshly hatched larvae were fed a standard fly food diet for defined periods of time (4, 12, 16, 20, or 24 hours), and growth and proliferation of neuroblasts, glia, and trachea were assayed in the brain (Fig 1). Fly food was supplemented with EdU (5-ethynyl-2′-deoxyuridine) to assay S-phase entry, and size was measured based on the average diameter of neuroblasts or total membrane surface area for glia and trachea (see Materials and Methods for details). After 4 hours of animal feeding, we observed that the 4 mushroom body (MB) neuroblasts (white arrows) and 1 lateral neuroblast were dividing based on their incorporation of EdU and generation of EdU-positive progeny (Fig 1A and 1F and S1A–S1C

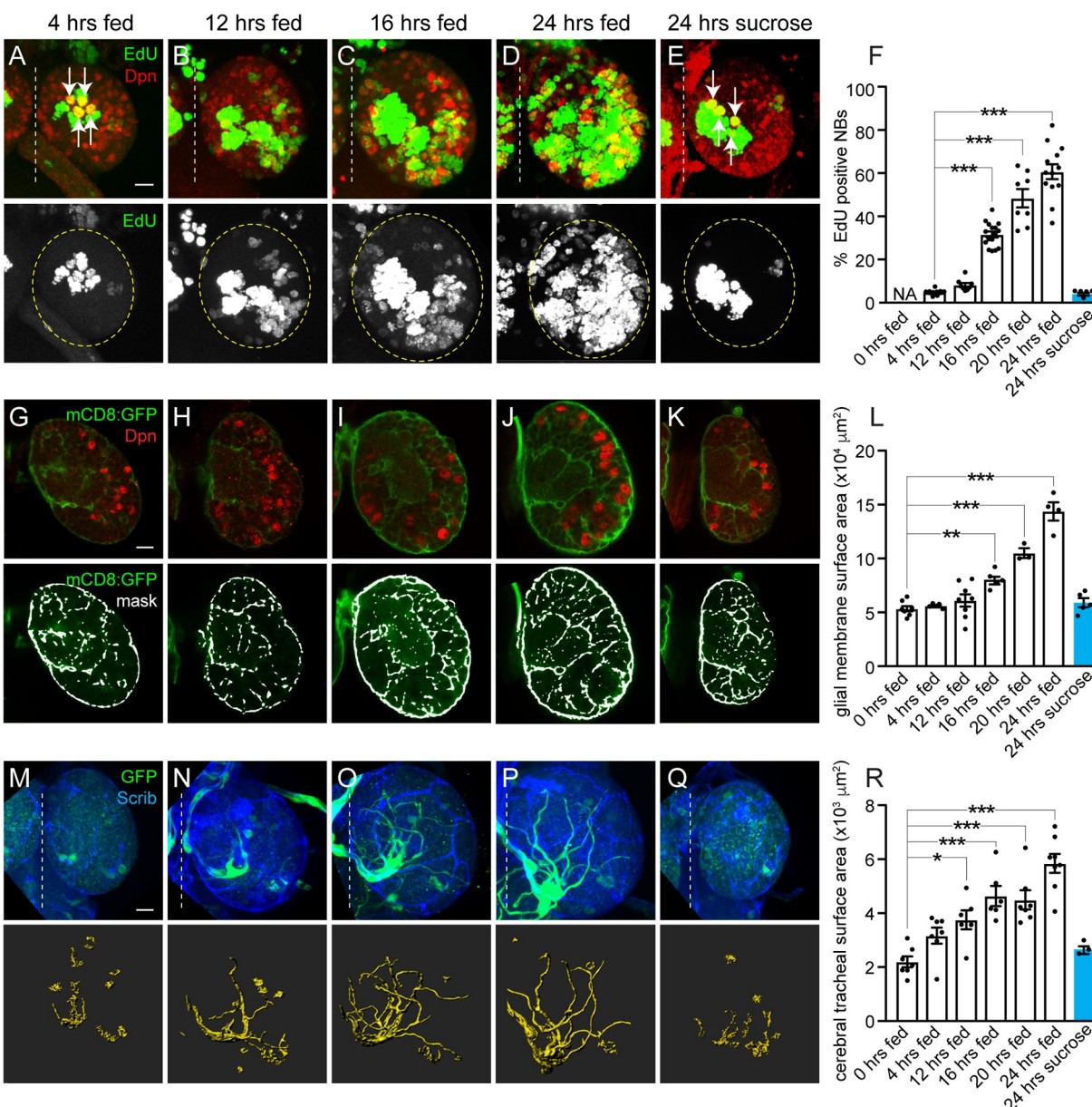

**Fig 1. NB reactivation from quiescence, as well as glial and tracheal growth, are nutrient regulated.** (A–E) Maximum intensity projections of single brain hemispheres. Top panels are colored overlays with single-channel grayscale images below. Brain hemispheres are outlined, and the dotted vertical line indicates the midline. Molecular markers are denoted within panels, and white arrows indicate MB NBs (A,E). (F) Quantification of EdU-positive NBs over time shown as scatter plots with dots representing individual brain hemispheres. Columns indicate mean and error bars indicate SEM in this and all subsequent figures. (G–L) Increases in glial membrane surface area over time in animals expressing mCD8:GFP using *repoGAL4*. (G–K) Single optical sections of brain hemispheres. Top panels are colored overlays, and bottom panels are single-channel images with mask overlays used in quantification (L) of total glial membrane surface area over time (see Materials and Methods). (M–R) Cerebral tracheal morphology over time in animals expressing GFP using *btlGAL4*. (M–Q) Maximum intensity projections of brain hemispheres with rendered trachea below and quantified in (R). (F,L,R) One-way ANOVA with Tukey post hoc analysis, $^*p < 0.05$, $^{**}p < 0.01$, $^{***}p < 0.001$. (A,G,M) Scale bar equals 10 µm in this and all subsequent figures. Genotypes of panels listed in S2 Table and data listed in S1 Data. *btl*, *breathless*; Dpn, Deadpan; EdU, 5-ethynyl-2′-deoxyuridine; GFP, green fluorescent protein; mCD8, membrane-targeted CD8 antigen; NB, neuroblast; *repo*, *reversed polarity*; Scrib, Scribble.

Fig). In contrast, the other central brain neuroblasts (approximately 100 per brain hemisphere), referred to as non-MB neuroblasts, failed to incorporate EdU during this time (Fig 1A and 1F and S1A–S1C Fig). EdU-positive MB and lateral neuroblasts were larger than the EdU-negative,

non-MB neuroblasts, but all expressed the HES1 orthologue, Deadpan (Dpn), a pan-neuroblast marker (S1B and S1C Fig). After 12 hours of feeding, a few non-MB neuroblasts incorporated EdU, and EdU incorporation correlated with increased neuroblast size (Fig 1B and 1F and S1F Fig). Over time, the fraction of EdU-positive, non-MB neuroblasts continued to increase, and at 24 hours, more than 60% of non-MB neuroblasts were EdU-positive (Fig 1C, 1D and 1F). Similar to earlier time points, EdU-positive non-MB neuroblasts were larger than EdU-negative non-MB neuroblasts (S1D and S1F Fig), consistent with previous reports [17, 21–23]. To confirm that increases in neuroblast size and S-phase entry are nutrient regulated, we fed animals a sucrose-only diet for 24 hours. Consistent with previous reports, no neuroblasts other than the 4 MB neuroblasts (white arrows) and the lateral neuroblast incorporated EdU (Fig 1E and 1F), and non-MB neuroblast size was reduced compared to both EdU-positive and negative non-MB neuroblasts from 24-hour–fed animals (S1E and S1F Fig). We conclude that neuroblasts reactivate from developmental quiescence in response to animal feeding and that reactivation occurs stepwise. Neuroblasts grow in size first and subsequently reenter S-phase and begin generating new progeny. This conclusion is in agreement with previously published work [17,21,22,25,26].

Next, we asked whether growth of other cell types within the brain is also nutrient regulated. Nutrient-dependent growth of glia in the ventral nerve cord (VNC) has been reported [23], yet it is not known whether growth of glia in the brain could play a role in regulating neuroblast reactivation from quiescence and contribute to the substantial increases in brain size observed after 24 hours of animal feeding (~1.76X, S1G Fig). After 24 hours of feeding, we found that approximately 45% of glia, identified based on expression of the homeodomain transcription factor reversed polarity (Repo), incorporated EdU (S1H and S1I Fig). This was unexpected and suggested that either new glia are being produced or that existing glia endoreplicate in response to feeding. To distinguish between these possibilities, we counted glial nuclei before animal feeding (0 hours fed, freshly hatched) and after 24 hours of animal feeding. We found approximately 95 Repo-positive glia per brain hemisphere before feeding and approximately 100 after feeding, suggesting that EdU incorporation is not followed by glial cell division (S1J Fig). Next, we expressed upstream activating sequence (*UAS*)–*double-parked* RNA interference (*dupRNAi*) in glia (using *repoGAL4*) to inhibit DNA replication and block endoreplication [27]. After 24 hours of feeding, essentially no *dupRNAi*-expressing glia were EdU-positive (<1%, $n = 5$ brain hemispheres), and the glial number was unchanged compared to controls, indicating that glia endoreplicate in response to feeding (S1K and S1L Fig). Next, we expressed a membrane-tagged green fluorescent protein (GFP) in glia (*repoGAL4, UAS*–membrane-targeted CD8 antigen [*mCD8*]*GFP*) to assay glial surface area over time. Between 12 and 24 hours of animal feeding, glial membrane surface area increased nearly 3-fold (Fig 1G–1J and 1L). To determine whether glial membrane growth and S-phase entry are nutrient regulated, animals were fed a sucrose-only diet. After 24 hours, no increases in glia membrane surface area or EdU-positive glia were found, demonstrating that glial membrane growth and S-phase entry are nutrient regulated (Fig 1K and 1L and S1I Fig), in agreement with previously published work [23]. We conclude that glia endoreplicate and their membrane surface area increases in response to dietary nutrients.

Next, we assayed growth of brain trachea in response to feeding. Trachea are a network of epithelial-derived tubules that supply oxygen and exchange gas throughout the animal. In the brain, during later larval stages, trachea extend along glia forming a perineuropilar tracheal plexus, analogous to cerebral vasculature in mammals [28]. During mammalian cortical development, intermediate neural progenitors divide near blood vessel branch points, suggesting that cerebral vasculature provides niche-like support, similar to *Drosophila* glia [29]. To determine whether tracheal morphogenesis in the *Drosophila* brain is nutrient regulated, we assayed tracheal growth over time in response to feeding. Before animal feeding, a single tracheal

branch enters the medial brain region [28]. After 24 hours of feeding, we observed 1 to 4 EdU-positive tracheal nuclei, located at the base of secondary branches, in each brain hemisphere (S1M and S1N Fig), and we found an overall 3-fold increase in tracheal surface area (Fig 1M–1P and 1R). Tracheal branching became more elaborate over time, with brain hemispheres being infiltrated from the inside out in a stereotypic pattern. In contrast, when animals were fed a sucrose-only diet, no EdU-positive tracheal nuclei were observed, and tracheal surface area and branching was reduced (Fig 1Q and 1R and S1N Fig). Together, we conclude that growth (S-phase entry and size) of neuroblasts, glia, and trachea is nutrient regulated. Moreover, growth of all 3 cell types occurs continuously and concomitantly, raising the possibility that nutrient-sensing pathways coordinate growth among different cell types.

## Nutrient-dependent growth of neuroblasts, glia, and trachea requires cell-autonomous and non-cell–autonomous activation of PI3-kinase

Increases in cell growth in response to dietary nutrients are typically due to increased PI3-kinase pathway activity, an evolutionarily conserved growth signaling pathway that activates TOR-kinase and other growth pathways [1,3,7,9,14]. To determine whether PI3-kinase is required for nutrient-dependent growth of neuroblasts, glia, and trachea, we expressed *UAS-Drosophila protein 60* (*dp60*) to reduce PI3-kinase activity cell autonomously using cell-type–specific GAL4 lines and assayed EdU incorporation and cell size or membrane surface area after 24 hours of feeding (Fig 2A and S2A–S2F Fig) [30]. When levels of PI3-kinase activity were reduced in neuroblasts (*worniu* [*wor*]*GAL4, UAS-dp60*), neuroblast EdU incorporation was reduced and neuroblast size reduced as reported previously (S2A and S2B Fig) [21,22,26]. When levels of PI3-kinase activity were reduced in all glia (*repoGAL4, UAS-dp60*), EdU incorporation was essentially absent in glia and membrane surface area reduced, consistent with work done in the ventral nerve cord (S2C and S2D Fig) [23]. Reduction of PI3-kinase activity in trachea (*btlGAL4, UAS-dp60*), resulted in reduced EdU incorporation and tracheal surface area and branching (S2E and S2F Fig). Therefore, PI3-kinase is required for cell autonomous nutrient-dependent growth of neuroblasts, glia, and trachea. This conclusion is in agreement with previously published work [21,22,23].

To determine whether PI3-kinase is also required to coordinate growth among neuroblasts, glia, and trachea in response to nutrition, levels of PI3-kinase activity were reduced in 1 cell type (neuroblasts, glia, or trachea) alone, and the growth of the other 2 cell types was assayed. When PI3-kinase levels were reduced in neuroblasts (*worGAL4, UAS-dp60*), a modest but significant reduction in glial membrane surface area was observed after 24 hours of feeding (Fig 2A–2C and 2E). A role for neuroblast PI3-kinase activation has been reported for chamber formation, a term coined to describe glia growth in response to neuroblast division rates [23]. To further support that neuroblast growth is required for glial growth, we expressed *raptorRNAi* in neuroblasts (*worGAL4, UAS-raptorRNAi*) to knock down TOR activity. Again, a significant reduction in glial membrane surface area was found after 24 hours of feeding (Fig 2D and 2E). Reductions in glial membrane surface area could be due to reductions in glia number because some *worGAL4*-expressing neuroblast lineages generate glia. Indeed, glia number was reduced compared with controls in *raptorRNAi* knockdown animals (*worGAL4, UAS-raptorRNAi*) but remained unchanged when PI3-kinase activity was reduced in neuroblasts (*worGAL4, UAS-dp60*) (S2G Fig). Next, we assayed tracheal surface area when levels of PI3-kinase activity were reduced in neuroblasts. No change in tracheal surface area was detected (S2H–S2J Fig). We conclude that activation of PI3-kinase signaling in neuroblasts is required for glial membrane growth, but not for growth of trachea.

Next, we reduced PI3-kinase activity in glia and assayed the growth of neuroblasts and trachea. When PI3-kinase levels were reduced in glia (*repoGAL4, UAS-dp60*), significant

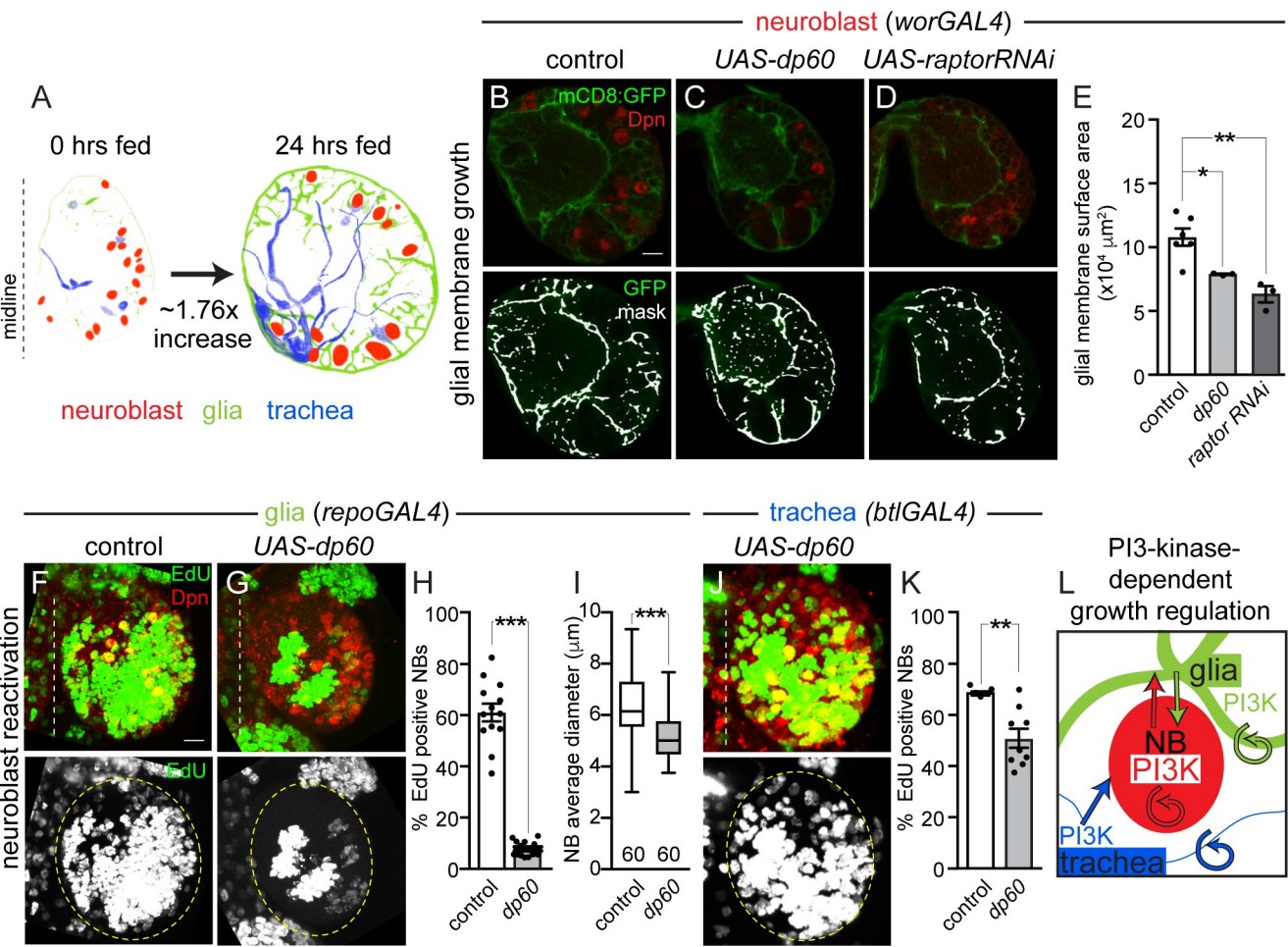

**Fig 2. Nutrient-dependent growth of NBs, glia, and trachea requires cell-autonomous and non-cell–autonomous PI3-kinase activation.** (A) Images of segmented brain hemispheres with cell types colored as indicated. (B–D) Single Z images of brain hemispheres. Top panels are colored overlays, and bottom panel are single-channel images overlaid with the mask used for quantification (E) (see Materials and Methods). (F,G,J) EdU-positive NBs after 24 hours of feeding. Maximum intensity projections of brain hemispheres from the indicated genotypes. Top panels are colored overlays, and single-channel grayscale images are below. Brain hemispheres are outlined, and the dotted vertical line to the left indicates the midline. Quantification of EdU-positive NBs (H,K) and NB size (I). (L) Summary of PI3-kinase growth regulation between different cell types in the brain. Circular arrows indicate requirement for autonomous PI3-kinase growth signaling, and straight arrows indicate nonautonomous growth signaling. (E) One-way ANOVA with Tukey post hoc analysis. (H,I,K) Student two-tailed $t$ test, $^*p < 0.05$, $^{**}p < 0.01$, $^{***}p < 0.001$, error bars, SEM. Genotypes of panels listed in S2 Table and data listed in S1 Data. *btl*, *breathless*; Dpn, Deadpan; *dp60*, *Drosophila protein 60*; EdU, 5-ethynyl-2′-deoxyuridine; GFP, green fluorescent protein; mCD8, membrane-targeted CD8 antigen; NB, neuroblast; PI3-kinase, phosphoinositide 3-kinase; *raptorRNAi*, *raptor* RNA interference; *repo*, *reversed polarity*; *UAS*, upstream activating sequence; *wor*, *worniu*.

reductions in both neuroblast EdU incorporation and neuroblast size were found after 24 hours of feeding (Fig 2F–2I). Yet, tracheal surface area remained relatively unchanged compared to controls (S2K and S2L Fig). Finally, we reduced PI3-kinase activity in trachea and assayed the growth of neuroblasts and glia. When PI3-kinase activity levels were reduced in trachea (*btlGAL4*, *UAS-dp60*), we found a modest but significant reduction in neuroblast EdU incorporation after 24 hours of feeding (Fig 2J and 2K) but no change in glia membrane surface area (S2M and S2N Fig). We conclude that activation of PI3-kinase signaling in glia and trachea both contribute to neuroblast reactivation, revealing that both cell types provide niche-like stem cell support. In addition, activation of PI3-kinase in neuroblasts is required for glial growth (Fig 2B–2E), but not tracheal growth (summary panel Fig 2L). This supports previous work showing that reductions in PI3-kinase levels in ventral cord neuroblasts leads to defects

in glial chamber formation [23]. Altogether, we conclude that PI3-kinase signaling functions in a cell-autonomous and nonautonomous manner to coordinate growth among different cell types within the developing *Drosophila* brain.

## Growth of cortex, subperineurial, and neuropil glia is nutrient regulated and PI3-kinase dependent

The glial population in the *Drosophila* brain is composed of several different subtypes, including cortex glia that ensheathe neuroblasts and their newborn neuron progeny, SPG that encapsulate the CNS and form part of the "blood-brain–like barrier", and neuropil glia that separate neuron cell bodies from their axon projections (S3A Fig). To better understand how PI3-kinase–dependent growth is coordinated between neuroblasts and glia, we assessed numbers and types of glia, based on location, before and after animal feeding. Before feeding, approximately 95 Repo-positive glia were found in each brain hemisphere. Of these, approximately 18 were cortex glia (19%), approximately 34 were neuropil glia (35%, which include ensheathing glia of the neuropil and astrocytes), and the rest, approximately 46%, were SPG and optic-lobe–associated glia combined. After 24 hours of feeding, glial number and subtype distribution based on location remained relatively similar (S3B Fig). We conclude that glial number and type are not specified by dietary nutrient uptake.

Next, we screened existing glial GAL4 lines to identify lines that drive GAL4 reporter expression specifically in different glial subsets in 24-hour–fed animals (S1 Table). We found that *NP0577GAL4* drove *UAS-histone* red fluorescent protein (*RFP*) reporter expression in all cortex glia, identified based on location and Repo coexpression, as well as in some other glia (S3C and S3D Fig and S1 Table). *MoodyGAL4* drove *UAS-histoneRFP* in most SPG, almost half of neuropil glia, but no other glia (S3E and S3F Fig and S1 Table). Of the 6 glial GAL4 lines screened, *NP0577GAL4* and *moodyGAL4* exhibited the most restricted and specific patterns of glia GAL4 expression in brains of 24-hour–fed animals. Therefore, we used these lines for subsequent analyses.

We expressed a membrane-tagged GFP in cortex glia (*NP0577GAL4, UAS-mCD8GFP*) to assay membrane growth in this glial subtype. After 24 hours of feeding, cortex glial membrane surface area increased nearly 3-fold, and cortex glia began to ensheathe neuroblasts with new glial membrane (Fig 3A and 3D). In contrast, animals fed a sucrose-only diet or a normal diet with reduced PI3-kinase levels in cortex glia (*NP0577GAL4, UAS-dp60*) showed no increases in cortex glia membrane surface area after 24 hours (Fig 3B–3D). Furthermore, we found no evidence of glial membrane ensheathment of neuroblasts under these conditions (Fig 3B and 3C, right panels). Next, we carried out a similar set of experiments in *moodyGAL4, UASmCD8GFP* animals. In animals fed a sucrose-only diet or those fed a normal diet but with reduced PI3-kinase levels in glia (*moodyGAL4, UAS-dp60*), we observed reductions in SPG and neuropil glial membrane surface area compared to animals fed a normal diet for 24 hours (Fig 3E–3H). Of note, neither SPG nor neuropil glia ensheathed neuroblasts (Fig 3E, right panel). We conclude that cortex glia, SPG, and neuropil glia require dietary nutrients and PI3-kinase activity for growth. Importantly, glial-subtype–specific GAL4 lines provide the tools necessary to further dissect nonautonomous growth regulation between neuroblasts and specific glia subtypes in response to nutrition and PI3-kinase activity levels. Previous work has reported similar results [23], yet glial subtype specific GAL4 lines were either not used or not verified for the developmental time assayed.

## Cortex glia are required to reactivate neuroblasts from developmental quiescence

We found that reduction of PI3-kinase activity in all glia inhibited neuroblast reactivation from developmental quiescence (refer back to Fig 2F–2I). Using glial-subtype–specific GAL4

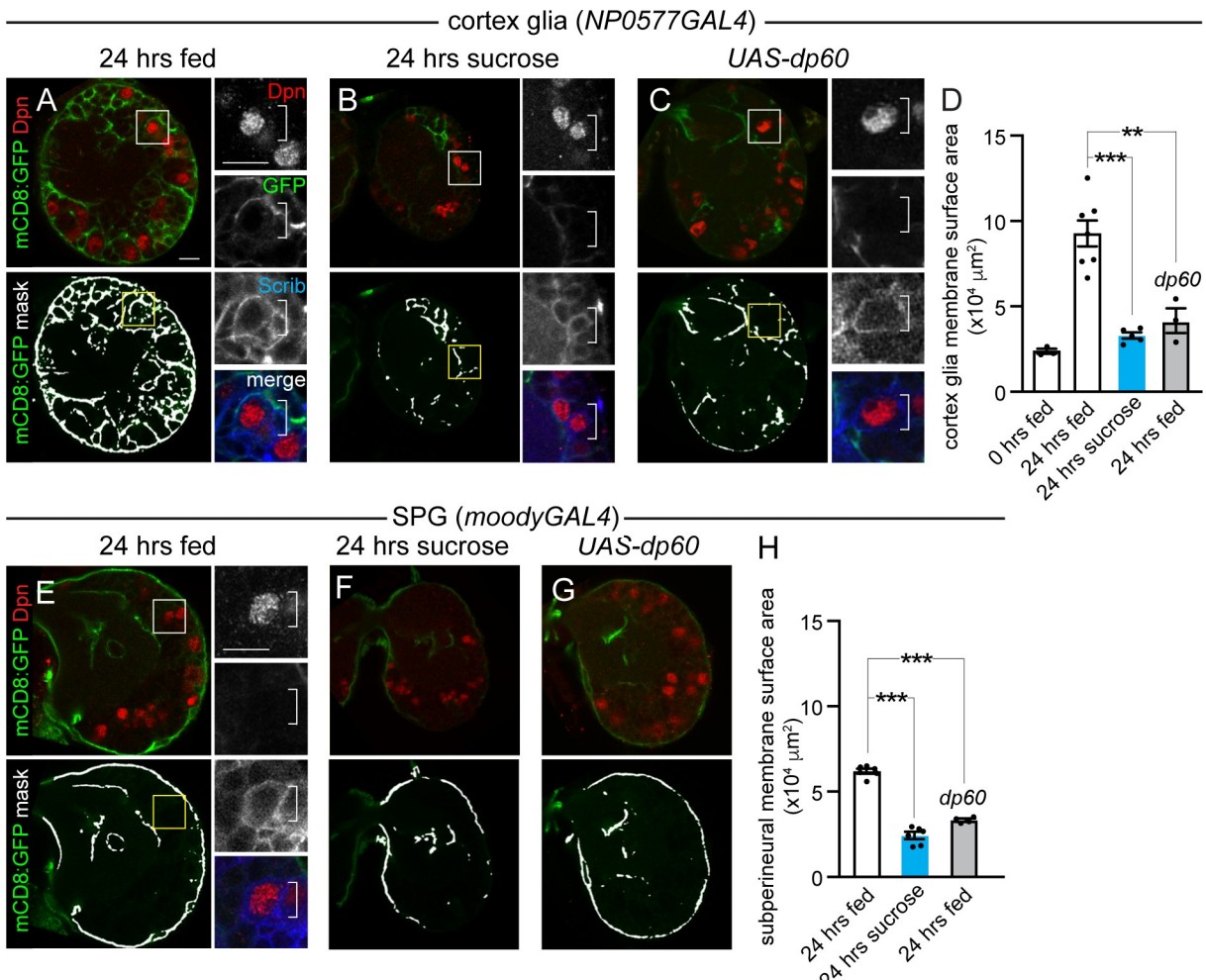

**Fig 3. Growth of cortex glia and SPG is nutrient regulated and PI3-kinase dependent.** (A-C) Cortex glial membrane morphology after 24 hours of feeding standard food (A and C) or sucrose (B). (A–C) Single Z planes of brain hemispheres. Top panels are colored overlays, and bottom panels are single-channel images with the mask overlays used for quantification of cortex glia membrane surface area (D). Boxed Dpn-positive NBs are shown at higher magnification to the right. (E–G) SPG and neuropil glial membrane morphology after 24 hours of feeding on standard food (E and G) or sucrose (F). (J–L) Single Z images of brain hemispheres. Top panels are colored overlays, and bottom panels show single-channel images with the mask overlay used for quantification of SPG and neuropil membrane surface area (H). Dpn-positive NBs denoted in the box shown at higher magnification to the right. One-way ANOVA with Tukey post hoc analysis, $^{**}p < 0.01$, $^{***}p < 0.001$. Error bars, SEM. Genotypes of panels listed in S2 Table and data listed in S1 Data. Dpn, Deadpan; *dp60*, *Drosophila protein 60*; GFP, green fluorescent protein; mCD8, membrane-targeted CD8 antigen; NB, neuroblast; PI3-kinase, phosphoinositide 3-kinase; Scrib, Scribble; SPG, subperineurial glia; *UAS*, upstream activating sequence.

lines, we reduced PI3-kinase levels in a glial-subtype–specific manner and assayed neuroblast EdU incorporation and size after 24 hours of feeding. When PI3-kinase activity was reduced in cortex glia (*NP0577GAL4*, *UAS-dp60*), we found that neuroblast EdU incorporation and size were reduced after 24 hours of animal feeding (Fig 4A, 4B, 4D and 4E). The same effect was observed when phosphatidylinositol (3,4,5)-trisphosphate (PIP$_3$) levels were reduced by overexpressing the lipid phosphatase, phosphatase and tensin homolog (Pten) (*NP0577GAL4*, *UAS-Pten*) (Fig 4D). However, when PI3-kinase activity levels were reduced in SPG/neuropil glia (*moodyGAL4*, *UAS-dp60*), no difference in neuroblast EdU incorporation was found compared to controls (Fig 4F–4H). Next, *UAS-dp60* was simultaneously expressed in both cortex glia and trachea. A significant reduction in neuroblast EdU incorporation was found

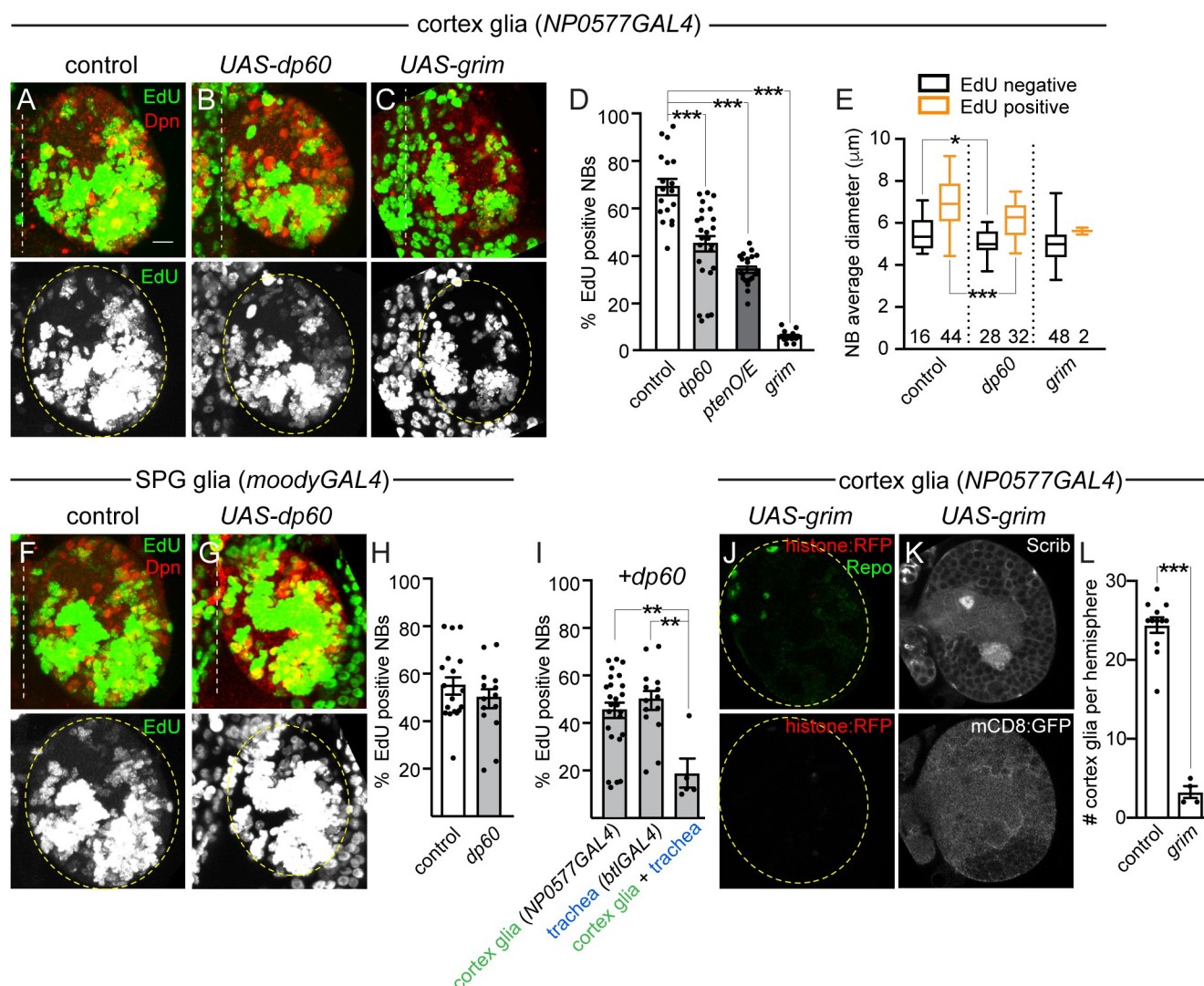

**Fig 4. Cortex glia are required for nutrient-dependent NB reactivation from quiescence.** (A–D) EdU-positive NBs in animals with genetic manipulations in cortex glia at 24 hours after feeding. (A–C) Maximum intensity projections of brain hemispheres. Top panels are colored overlays, and bottom panels show single-channel grayscale images. Brain hemispheres are outlined, and the dotted vertical lines indicate the midline. Graphs show quantification of EdU-positive NBs per brain hemisphere (D) and NB size based on EdU incorporation (E). Numbers in E indicate the number of NBs analyzed. (F–H) EdU-positive NBs in animals with genetic manipulations in SPG and neuropil glia at 24 hours after feeding. (F–G) Maximum intensity projections of brain hemispheres. Panels are colored overlays (top), with single-channel grayscale images below. Brain hemispheres are outlined, and dotted vertical lines indicate the midline. Quantification of EdU-positive NBs per brain hemisphere (H). (I) Quantification of EdU-positive NBs per brain hemisphere of indicated genotypes. (J) Single Z-plane image, top panel is a colored overlay with single-channel grayscale image below, brain hemisphere outlined. (K) Two (top and bottom) grayscale images of same Z-plane, with quantification of cortex glia number following genetic ablation quantified in (L). (D,I) One-way ANOVA with Tukey post hoc analysis and (E,H,L) Student two-tailed $t$ test, $^*p < 0.05$, $^{**}p < 0.01$, $^{***}p < 0.001$. Genotypes of panels listed in S2 Table and data listed in S1 Data. *btl*, *breathless*; Dpn, Deadpan; *dp60*, *Drosophila protein 60*; EdU, 5-ethynyl-2′-deoxyuridine; mCD8, membrane-targeted CD8 antigen; NB, neuroblast; O/E, overexpression; Pten, phosphatase and tensin homolog; Repo, reversed polarity; *RFP*, red fluorescent protein; Scrib, Scribble; SPG, subperineurial glia; *UAS*, upstream activating sequence.

compared to animals expressing *UAS-dp60* alone in either cortex glia (*NP0577GAL4*) or trachea (*btlGAL4*) (Fig 4I). Finally, to confirm that cortex glia are indeed required for neuroblast reactivation, the proapoptotic gene *grim* was expressed (*NP0577GAL4, UAS-grim)* to ablate cortex glia genetically (Fig 4J–4L). After 24 hours of feeding, neuroblast EdU incorporation was essentially absent, and neuroblast size was reduced compared with controls (Fig 4C–4E). We conclude that neuroblast growth and reactivation from quiescence requires activation of

PI3-kinase in cortex glia, the glial subtype that ensheathes neuroblasts and their newborn progeny, but not SPG or neuropil glia as reported in other work [23]. Furthermore, activation of PI3-kinase growth signaling in both trachea and cortex glia together promote neuroblast reactivation from developmental quiescence.

## Dilp-2 regulates neuroblast reactivation and cortex glia membrane growth, but not growth of trachea

PI3-kinase is activated in response to feeding, after any one of the 7 *Drosophila* insulin-like peptides (Dilps1–7) bind to and activate the single insulin-like tyrosine kinase receptor (InR). To date, neuroblast reactivation has been reported to fail in *dilp6* single mutants, *dilp2dilp3* double mutants, *dilp2dilp3dilp5* triple mutants, *dilp2dilp3dilp5dilp6* quadruple mutants, and *dilp1dilp2dilp3dilp4dilp5* quintuple mutants [21,22], however a systematic analysis of each of the single dilp mutants is still lacking as well as phenotypic quantification associated with any *dilp* mutant analysis. To better understand how PI3-kinase coordinates growth among different cell types within the developing brain, we assayed neuroblast EdU incorporation in each of the 7 single *dilp* null mutants. Compared to controls and other *dilp* mutants, neuroblast EdU incorporation in the brain was reduced in *dilp1*, *dilp2*, and *dilp7* null mutants after 24 hours of feeding (Fig 5A). Of these, *dilp2* mutants displayed the most severe and penetrant reductions in neuroblast EdU incorporation and neuroblast size (Fig 5A–5D and S4A Fig). To confirm that reduced neuroblast EdU incorporation was due to the absence of *dilp2* and not a second background mutation, we assayed EdU incorporation in animals transheterozygous for the *dilp2* null allele over a small deficiency that removes *dilp2* and the neighboring *dilp3* locus. The number of EdU-positive neuroblasts after 24 hours of feeding in transheterozygous larvae was indistinguishable from *dilp2*[1] homozygotes (Fig 5A). We conclude that Dilp-1, Dilp-2, and Dilp-7 regulate neuroblast reactivation from quiescence in the brain.

Given the prominent role of Dilp-2 in neuroblast growth and reactivation in the brain, we next asked whether Dilp-2 is also required for cortex glial and tracheal growth in response to animal feeding. Compared to controls, cortex glial membrane surface area was reduced by half in *dilp2* mutants and in animals transheterozygous for the *dilp2* null allele over the *dilp2*, *dilp3* deficiency after 24 hours of feeding (Fig 5E–5G). In contrast, tracheal surface area remained unchanged (Fig 5H–5J). Next, we assayed neuroblast reactivation and cortex glial membrane surface area at 48 hours ALH (after larval hatching). No difference in number of EdU-positive neuroblasts or glial cortex membrane surface was observed between controls and *dilp2* mutants (S4B and S4C Fig). Because *dilp2* mutants reportedly delay development anywhere from 8 to 16 hours from egg to adult [45], we assayed mouth hook morphology as a proxy for larval stage. No difference between controls and *dilp2* mutants was observed at 24 or 48 hours ALH, suggesting that *dilp2* null mutants are not developmentally delayed during early larval stages (S4D and S4E Fig). We conclude that Dilp-2 is required for cortex glial membrane growth, but not tracheal growth. Furthermore, because Dilp-2 is also required for neuroblast growth and reactivation, it suggests that growth coordination between cortex glia and neuroblasts is Dilp-2 dependent.

## Dilp-2 regulates CNS Dilp-6 protein levels, but not *dilp6* transcript levels

Somewhat surprisingly, neuroblast EdU incorporation in brains of *dilp6* null-mutant animals was not different compared with controls, *dilp3*, *dilp4*, or *dip5* mutants, nor was neuroblast size affected by loss of *dilp6* (Fig 5A, 5B and 5D and S4A Fig). This was unexpected because Dilp-6 is reported to be synthesized and secreted from SPG in response to animal feeding, leading to PI3-kinase activation and reactivation of adjacent quiescent neuroblasts in the

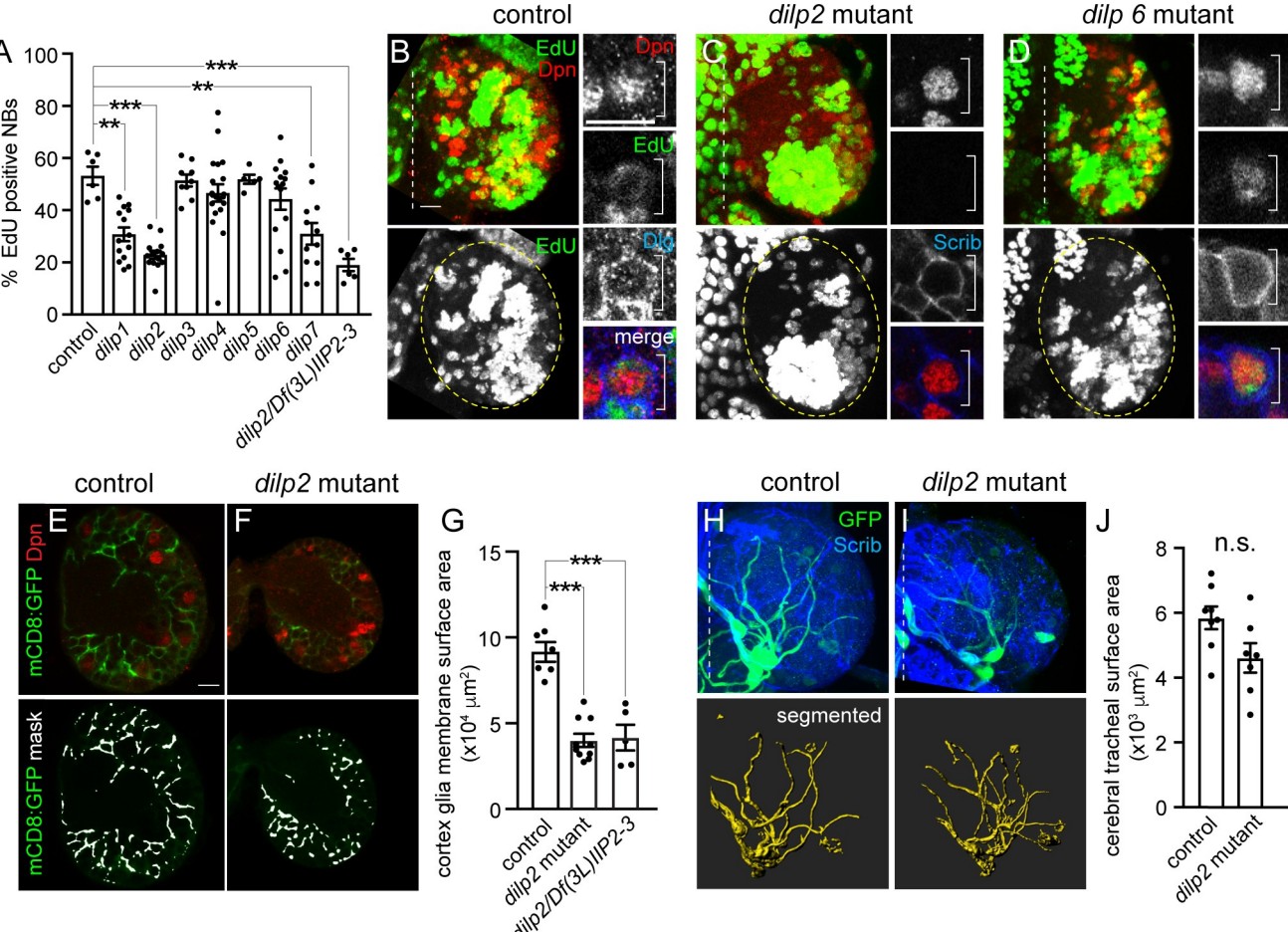

**Fig 5. Dilp-2 regulates NB reactivation and cortex glial membrane growth.** (A–D) EdU-positive NBs after 24 hours of feeding in *dilp* single-null–mutant animals with quantification (A) shown as a scatter plot with dots indicating brain hemispheres, bars indicating means, and error bars SEM. (B–D) Maximum intensity projections of brain hemispheres. Top panels are colored overlays with single-channel grayscale images below. Brain hemispheres are outlined, and dotted vertical lines indicate the midline. (E–G) Cortex glial membrane morphology after 24 hours of feeding in controls and *dilp2* mutants. (E,F) Single Z images of brain hemispheres. Top panels are colored overlays, and bottom panels are single-channel images with the mask overlays used for quantification in (G). (H–J) Cerebral tracheal morphology after 24 hours of feeding in controls and *dilp2* mutants. (H,I) Maximum intensity projections of brain hemispheres with rendered trachea below and quantified in (J). (A,G) One-way ANOVA with Tukey post hoc analysis and (J) Student two-tailed *t* test, $^*p < 0.05$, $^{**}p < 0.01$, $^{***}p < 0.001$. Genotypes of panels listed in S2 Table and data listed in S1 Data. Dilp, *Drosophila* insulin-like peptide; Dlg, discs-large; Dpn, Deadpan; EdU, 5-ethynyl-2′-deoxyuridine; GFP, green fluorescent protein; mCD8, membrane-targeted mCD8 antigen; NB, neuroblast; n.s., not significant; Scrib, Scribble.

ventral nerve cord (VNC) [21–23]. However, neuroblasts located within different CNS regions, brain versus VNC, could have different requirements for reactivation based on neuroblast intrinsic differences and/or differences in tissue architecture. To determine whether Dilp-6 regulates growth of other cell types within the brain, we assayed glial and tracheal surface area in *dilp6* mutants after 24 hours of feeding. No differences were observed compared to controls (Fig 6A–6F). To determine whether Dilp-2 masks Dilp-6 function in the brain, we assayed neuroblast EdU incorporation and size in *dilp2*, *dilp6* double-mutant animals. After 24 hours of feeding, neuroblast EdU incorporation was reduced in *dilp2*, *dilp6* double mutants (approximately 15% EdU-positive neuroblasts) compared with *dilp2* single mutants (approximately 22% EdU-positive neuroblasts); however, this reduction was not statistically different (S5A and S5B Fig). Next, we assayed *dilp2* and *dilp6* transcript levels (Fig 6G and 6H), and

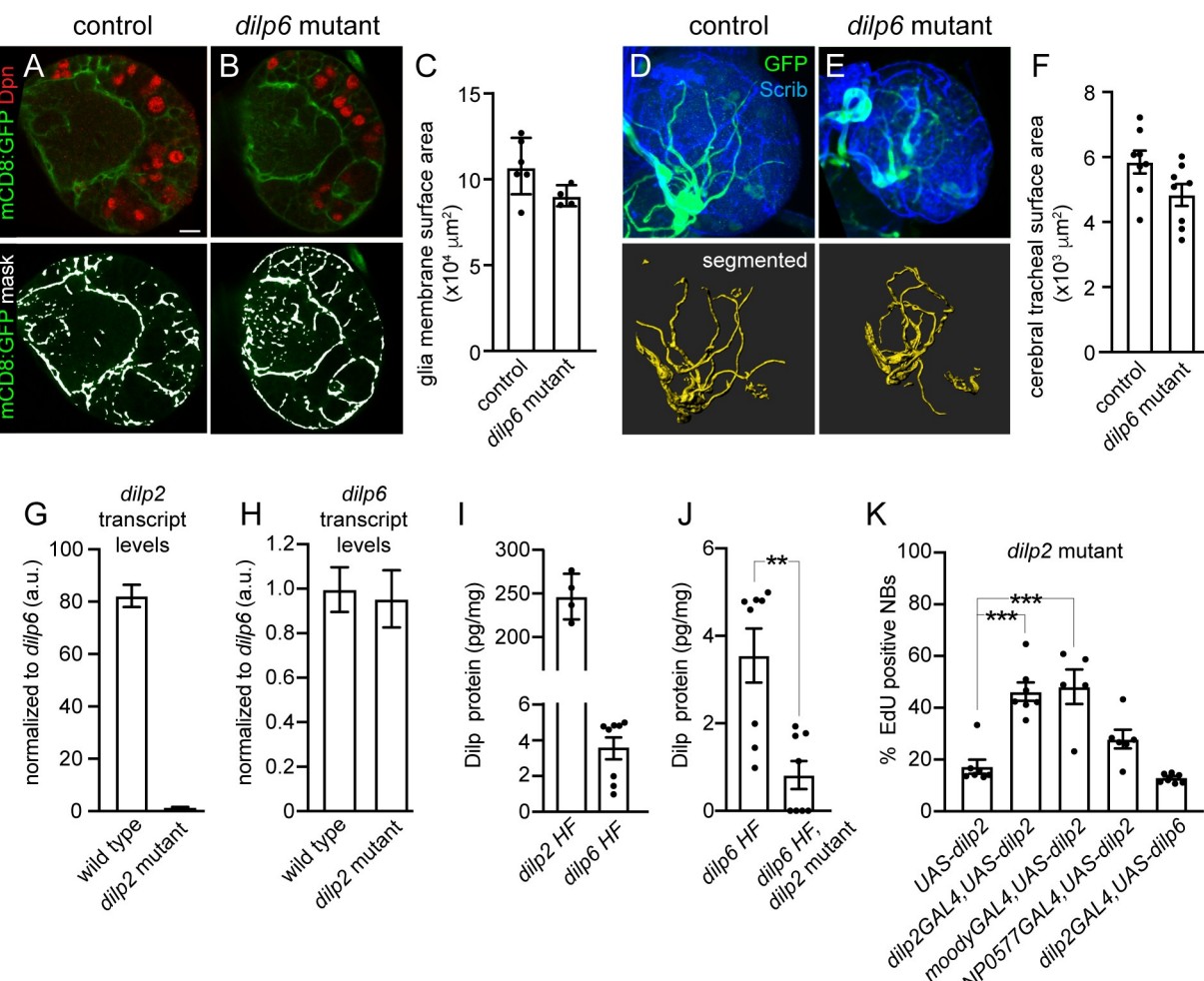

**Fig 6. Dilp-2 regulates Dilp-6 CNS protein levels.** (A–C) Glial membrane morphology after 24 hours of feeding in controls and *dilp6* mutants. (A,B) Single Z images of brain hemispheres. Panels are colored overlays (top) and single-channel images with mask overlays (bottom) used for quantification of total glial membrane surface (C). (D–F) Cerebral tracheal morphology after 24 hours of feeding in controls and *dilp6* mutants. (D,E) Maximum intensity projections of brain hemispheres with rendered trachea below and quantified in (F). (G,H) RT-qPCR analysis of *dilp2* and *dilp6* transcript levels in the CNS of wild-type and *dilp2* mutant animals after 24 hours of feeding. Transcript levels of *dilp2* and *dilp6* are normalized to *Gapdh1* and then to *dilp6* levels in wild-type animals. (I,J) Brain Dilp-2 and Dilp-6 protein levels, normalized to total brain protein, in wild-type and *dilp2* mutant animals after 72 hours of feeding. (K) Quantification of EdU-positive NBs in the indicated genotypes after 24 hours of feeding. (C,F,J) Student two-tailed *t* test; (K) one-way ANOVA with Tukey post hoc analysis and $^{**}p < 0.01$, $^{***}p < 0.001$. Genotypes of panels listed in S2 Table and data listed in S1 Data. a.u., arbitrary unit; CNS, central nervous system; Dilp, *Drosophila* insulin-like peptide; Dpn, Deadpan; EdU, 5-ethynyl-2′-deoxyuridine; *Gapdh1*, Glyceraldehyde-3-phosphate dehydrogenase; GFP, green fluorescent protein; HF, HA-FLAG tag; mCD8, membrane-targeted CD8 antigen; NB, neuroblast; RT-qPCR, real-time qPCR; Scrib, Scribble; *UAS*, upstream activating sequence.

endogenous Dilp-2 and Dilp-6 protein levels (Fig 6I) in the CNS of wild-type animals after feeding. We found that Dilp-2 transcript and protein levels were approximately 80 times higher than Dilp-6 transcript and protein levels, consistent with the notion that Dilp-2 is the predominant Dilp in the central brain (Fig 6G–6I). Furthermore, in *dilp2* mutants, we found that *dilp6* transcript levels were not different from wild-type control levels, although Dilp-6 protein levels were significantly reduced compared to control animals (Fig 6H and 6J). We conclude that Dilp-2 is the primary Dilp regulating PI3-kinase–dependent growth of neuroblasts, glia, and trachea in the brain and that Dilp-2 directly or indirectly regulates Dilp-6 protein levels.

Next, we carried out a series of rescue experiments in *dilp2* mutants to identify the cell source of Dilp-2 required for neuroblast reactivation. When *UAS-dilp2* was expressed in insulin-producing cells (IPCs, *dilp2 GAL4*), neuroblast EdU incorporation was rescued in *dilp2* mutants (Fig 6K). Similarly, expression of *UAS-dilp2* in SPG (*moodyGAL4*) also rescued neuroblast EdU incorporation in *dilp2* mutants. Because Dilp-2 is normally expressed in the IPCs [42,43] and not BBB (blood–brain barrier) glia, this suggests that Dilp-2 is required in IPCs for neuroblast reactivation in the brain. In contrast, expression of *UAS-dilp6* in IPCs did not rescue neuroblast EdU incorporation in *dilp2* mutants. We conclude that Dilp-2 is required in the IPCs for neuroblast reactivation from developmental quiescence in the brain hemispheres.

## Coordination of growth between neuroblasts and cortex glia promotes formation of a selective membrane barrier for niche stem cell support

During later stages of larval development, neuroblasts and their progeny reside within characteristic glial membrane-bound pockets (Fig 7A and 7B). One or two cortex glia that lie nearby or adjacent to neuroblasts and their progeny provide the membrane that constitutes each pocket. We injected a fluorescently conjugated 10-kDa dextran directly into the brain to test the permeability of glia membrane-bound pockets. We found fluorescent dextran colocalized with cortex glia membrane along the outside of each pocket, but we did not find fluorescence within the pocket (Fig 7A). This suggests that cortex glia form a membrane barrier that selectively regulates passage of factors based on size.

Next, we investigated whether Dilp-2–mediated PI3-kinase activation in neuroblasts and glia is required for glia pocket formation. First, we measured the fraction of neuroblast membrane, marked by endogenous expression of Scribble (Scrib), in contact with glia membrane, marked by transgenic expression of *UAS-mCD8:GFP* (% neuroblast membrane with glial contact; see Materials and Methods) over time in control animals (Fig 7B, schematic at bottom). From 0 hour freshly hatched stages until 8 hours after feeding, we found that the fraction of neuroblast membrane in contact with glial membrane increased approximately 3.5-fold (Fig 7C–7F). At 24 hours after feeding, glia ensheathe approximately 44% of the neuroblast membrane, and at 72 hours after feeding, they ensheathe approximately 72% (Fig 7B–7D and 7G). Next, we assessed the temporal relationship between neuroblast–glial membrane contact and changes in neuroblast size. We found that neuroblast size began to increase 16 hours after feeding, after the fraction of neuroblast–glial contact increased (Fig 7C–7G). Next, we measured the fraction of neuroblast membrane in contact with glia membrane in 24-hour–fed *dilp2* mutants (Fig 7I and 7K). We found significant reductions in neuroblast membrane and glia membrane contact that correlated with reductions in neuroblast size. A similar result was found when levels of PI3-kinase activity were reduced in glia, but not in neuroblasts (Fig 7H, 7J and 7K). In neuroblasts, when PI3-kinase activity levels were reduced, neuroblast size remained reduced compared to controls, but increases in neuroblast membrane in contact with glial membrane were found. We conclude that Dilp-2 mediated PI3-kinase activation is required to initiate glial pocket formation. Importantly, increases in neuroblast membrane with glia contact precede increases in neuroblast size, consistent with the notion that cortex glia membrane contact triggers neuroblast growth and reactivation.

## Discussion

Here, we report that different cell types in the brain utilize the same evolutionarily conserved cell signaling pathway, PI3-kinase, to promote growth in response to dietary nutrients, in agreement with work published previously [17,21–23]. PI3-kinase is a lipid kinase that converts $PIP_2$ to $PIP_3$ at the plasma membrane when active, leading to downstream Akt/PKB

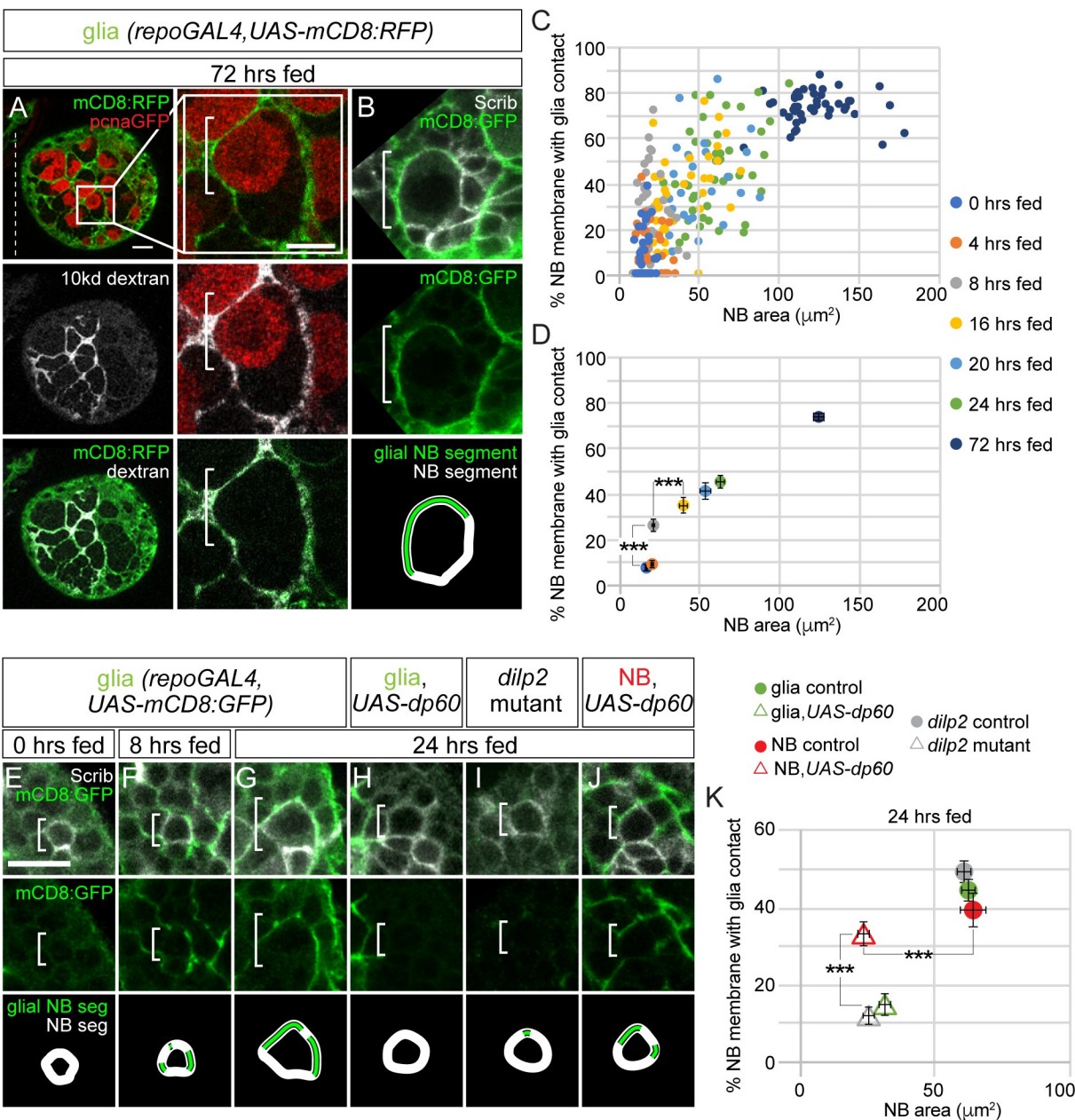

**Fig 7. Cortex glia ensheathe NBs and their progeny and provide a niche-like barrier function.** (A) Top panel, colored overlay with RFP marking glial membranes and GFP marking *pcnaGFP*-expressing NBs. Middle panel, single-channel grayscale image of 10-kDa dextran. Bottom panel, colored overlay with 10-kDa dextran and RFP marking glial membranes. To the right, high-magnification images of the boxed NB, with double-labeled glial membrane in a 72-hour–fed animal after dextran injection (see Materials and Methods). (B) Cortex glia ensheathe NBs (white bracket) and their newborn progeny, which express high levels of Scrib. Top panel, colored overlay. Middle panel, single-channel image with GFP marking glial membranes. Bottom panel is a depiction of the glial and NB segments used for quantification in (C,D,K) (see Materials and Methods). (C,D) Percentage of NB membrane in contact with glial contact over time. Colored circles in (C) indicate individual NBs, and colored circles in (D) indicate averages for the indicated time points. Both are plotted relative to NB area. (E–G) Increasing surface contact between NBs and glia membrane over time. Single Z-plane of representative NBs at different time points after feeding. Top panels are colored overlays, middles panels are single-channel images, and bottom panels depict the glial NB segment and NB segment used in quantification (C–D). (H–K) Top panel colored overlay, middle panel single-channel image, and bottom panel glial NB segment and NB segment used in quantification of animals at 24 hours of feeding with genotypes listed above. (D,K) One-way ANOVA with Tukey post hoc analysis, ***$p < 0.001$. Genotypes of panels listed in S2 Table and data listed in S1 Data. Dilp, *Drosophila* insulin-like peptide; GFP, green fluorescent protein; mCD8, membrane-targeted CD8 antigen; NB, neuroblast; *pcna*, proliferating cell nuclear antigen; *repo*, *reversed polarity*; RFP, red fluorescent protein; Scrib, Scribble; *UAS*, upstream activating sequence.

(Protein Kinase B) activation and nuclear exclusion of forkhead box, subgroup O (Foxo), a transcription factor and negative regulator of growth [31–34]. We found that active PI3-kinase not only regulates cell growth autonomously in the brain but also regulates growth of other cell types in a nonautonomous manner. For example, glial growth in the central brain is reduced when PI3-kinase levels are reduced in neuroblasts, a result reported in the ventral nerve cord as well [23]. The molecular basis for this phenotype is not yet known, but reactivated and proliferating neuroblasts with active PI3-kinase may express or secrete factors on their cell surface that promote glial growth nonautonomously. These could include PDGF- and VEGFR-receptor related (PVR) (Platelet-derived growth factor receptor [PDGFR] orthologue) ligands, secreted growth factors, or components of the Hippo or Notch cell signaling pathways that would not normally be expressed in quiescent neuroblasts having low to no PI3-kinase activity [35–37]. We also found that PI3-kinase–dependent growth of cortex glia is required for neuroblast reactivation and that cortex glia membrane ensheathes neuroblasts before they increase in size. Because the time points included in this analysis are early and at short intervals (up to 24 hours ALH at the time when neuroblasts reactivate), we can differentiate order of progressive cell growth phenotypes based on morphology and molecular markers. The glial process that makes first contact with a neuroblast could deliver signaling molecules to stimulate neuroblast reactivation from quiescence, similar to cytonemes or nanotubules, or simply supply a foreign membrane that stimulates growth [38,39]. We also found that neuroblasts rely on PI3-kinase-dependent growth of trachea for reactivation. Trachea allow for oxygen exchange, but they are also important sources of cell signaling molecules, including EGF-R and FGF-R. Whether neuroblast quiescence versus proliferation decisions are regulated by oxygen levels has yet to be determined, but quiescent adult stem cells in other organisms are thought to reside within hypoxic microenvironments [40,41].

Of all the Dilps, we found that Dilp-2 plays a predominant role in mediating PI3-kinase–dependent growth of neuroblasts and glia as well as neuroblast reactivation. This was not expected based on previous reports; however, neuroblasts with different intrinsic programs located within different regions of the CNS (brain versus VNC) could reactivate in a different manner. In the brain, Dilp-2 is synthesized and secreted in a dietary-nutrient–dependent manner from 14 neurosecretory neurons, the insulin-producing cells (IPCs), located along the anterior midline [42,43]. Axons of the IPCs terminate on the dorsal vessel, which pumps hemolymph, and thus Dilp-2, systemically throughout the body [43]. Dilp-2 is also released locally in the brain through IPC connections with other neurosecretory neurons, including the centrally located Dilp-recruiting neurons (DRNs) [44]. Whether local or systemic Dilp-2 activates PI3-kinase in the different brain cell types remains an open question. Once released, Dilp-2 could activate PI3-kinase in the different cell types in a simultaneous or sequential manner. Furthermore, Dilp-2 could stimulate synthesis and secretion of other Dilps and other factors that could promote autonomous or nonautonomous growth, either locally within the brain or systemically. Such factors may include Dilp-6, which is reported to be expressed in glia, and other Dilps that are expressed in neurosecretory neurons including the IPCs [21,22,42,43,45].

In the brain, different cell types coordinate growth in response to dietary nutrients. Growth among different tissue types is also coordinated and relies on systemic factors, including the relaxin-like hormone, Dilp-8 [46–48]. For example, in response to tissue damage, Dilp-8 is expressed and secreted systemically, inducing developmental arrest and allowing time for tissue damage to self-repair. Once repair is complete, Dilp-8 levels decrease, development resumes, and adults emerge with appropriately sized and proportioned appendages [46–50]. Systemic growth is also regulated by 20-hydroxyecdysone, an ecdysteroid hormone that promotes growth globally and couples increases in cell and tissue size with developmental

progression [2,5,6]. Dietary nutrient conditions play a key role in both ecdysone production and developmental timing, as PI3-kinase regulates growth of the prothoracic gland that synthesizes and secretes ecdysone. Accumulated nutrient biomass is monitored by a "critical weight" check point in development: before critical weight, development is delayed if dietary nutrients are limited, but once animals reach "critical weight," commitment to metamorphosis commences without delay regardless of dietary nutrients [1–6]. It will be important to determine whether coordinated cell growth occurs during tissue repair in periods of limited nutrient availability, and whether coordinated cell growth continues after "critical weight" is reached.

In mammals, cell growth decisions in response to nutrient availability are regulated by PI3-kinase and TOR-kinase signaling as well. However, how growth decisions are made and whether they are coordinated in response to dietary nutrients among stem cells within their local microenvironments is not well understood. Adult mammals, unlike flies, maintain a population of proliferative neural stem cells in the brain, some of which reside within rosette-like structures located along the walls of the lateral ventricle [51–52]. Multiciliated ependymal cells form the rosette with the stem cell and its primary cilium at the center [53]. Both ependymal cells and stem cells receive signals through their cilia, which project into the ventricles filled with cerebral spinal fluid (CSF). The choroid plexus produces CSF and with it, a host of factors, including insulin-like growth factor (IGF) [54]. Whether IGF regulates levels of PI3-kinase signaling in neural stem cells within their niche remains unclear, but loss of the rosette-like pinwheels correlates with decreased adult neurogenesis and premature progenitor differentiation [53]. In the future, it will be interesting to determine whether dietary nutrients regulate choroid plexus IGF levels and how stem cells and their niche respond to IGF during homeostasis and aging.

There have been a number of previous studies and landmark papers demonstrating the importance of dietary nutrients with subsequent activation of PI3-kinase growth signaling in regulation of neuroblast growth and reactivation from developmental quiescence, as well as in the expansion of glial membranes and formation of glial chambers [17,21–23,25,26]. Results reported here add to and expand on work published by others in the following important ways. Here, we document at the whole-tissue level changes in cell size and S-phase entry among three different cell types (neuroblasts, glia, and trachea) in the brain in response to nutrition. In contrast to previous work which measured fluorescent pixel intensities as a read-out for glial growth and cell type organization, we quantified total membrane surface of both glia and trachea in entire brain hemispheres and correlated these changes with changes in neuroblast growth and reactivation. Because of this holistic and very granular analysis over short time intervals, we were able to precisely document glial and tracheal growth in relation to neuroblast reactivation and determine that glia membranes contact neuroblasts before neuroblast S-phase entry, suggesting that glial membrane contact may be an important trigger for quiescence reactivation. We also found that trachea, like glia, provide neuroblast niche function. Furthermore, as stated previously, we determined that growth of these three cell types occurs concurrently and is coordinated. The idea of coordinated growth or non-autonomous growth regulation has been proposed previously, where it was reported that PI3-kinase activation in neuroblasts promotes glial chamber formation in the ventral nerve cord [23]. Here we further develop this idea and find that, in the brain, neuroblast PI3-kinase activity regulates the growth of glial membrane and promotes early glial membrane contact with quiescent neuroblasts. Furthermore, we find that PI3-kinase dependent growth of the cortex glia, which contact quiescent neuroblasts, is required for neuroblast reactivation in the brain. When cortex glia are ablated, neuroblasts fail to reactivate, a finding not published previously. Moreover, we found that when levels of PI3-kinase are reduced in SPG, no defect in neuroblast reactivation was

found. Together, with the fact the IPC Dilp2 is required, but not SPG Dilp6, our data suggests that neuroblast reactivation in the brain may be regulated differently than neuroblast reactivation in the ventral nerve cord.

## Materials and methods

### *Drosophila* stocks

Genotypes of fly stocks used in this study and their source are listed in S2 and S3 Tables.

### *Drosophila* maintenance and feeding methods

Larvae were collected immediately after hatching and placed on either Bloomington fly food diet or in a sucrose-only solution. Freshly hatched larvae were maintained at 25˚C on a 12-hour light/dark cycle for defined periods of time. Fly food or sucrose solution were supplemented with 0.1 mM EdU, and EdU incorporation was detected using commercially available Click-iT EdU Proliferation Kits for Imaging (Thermo Fisher Scientific, Waltham, MA, USA) as described previously [26].

### Immunofluorescence, dextran injections, and confocal imaging

Larval brains were fixed and stained as previously described [26]. In brief, early staged larval brains were dissected and fixed in 4% EM grade formaldehyde in PEM (Pipes, EGTA, magnesium chloride) buffer with 0.1% Triton-X for 20 mins. Tissues were washed in 1× PBT with 0.1% Triton-100 and blocked overnight at 4˚C in 1X PBT with 10% normal goat serum. Primary antibodies used in this study include chicken anti-GFP (1:500; Abcam, Cambridge, MA, USA), rat anti-Dpn (1:100; Abcam), rabbit anti-Scribble (1:1,000), mouse anti-Repo (1:5; DSHB, 8D12). Primary antibodies were detected using Alexa Fluor-conjugated secondary antibodies. For dextran dye experiments, 10-kDa dextran conjugated to Alexa Fluor 647 was injected into L3 brains maintained in culture using micromanipulators. Brains were then fixed and stained as described above. Brains were imaged using a Leica SP8 laser scanning confocal microscope (Leica, Wetzlar, Germany) equipped with a 63×, 1.4 NA oil-immersion objective. For glia and trachea surface measurements, Z stacks were acquired at 0.5-μm steps using same confocal settings. All other confocal data were collected at 1.0-μm steps. Images were processed using the Imaris and Fiji software packages.

### Cell counts and quantification of glia and tracheal membrane surface area

For neuroblast EdU quantification, the number of EdU-positive, Dpn-positive neuroblasts were counted using the "cell counter" plugin in Fiji and divided by the total number of Dpn-positive neuroblasts. For quantification of glia, the total number of Repo-positive nuclei were counted in each brain hemisphere and cortex, SPG, and neuropil glia distinguished based on their location and morphology. Neuroblast diameter was calculated based on the average length of 2 perpendicular lines drawn through center of the neuroblast at its widest point. To quantify glial and trachea surface area, GFP-expressing membranes were segmented and analyzed using the *Surface* module in Imaris 9.0.2 (Bitplane). Imaris Surface generates a volumetric surface object based on fluorescence intensity and creates a 3D surface mesh based on voxel intensity. After segmentation, a mask was generated and confirmed by manual visual inspection using Imaris slice mode. All confocal images used for surface area measurements were acquired using the same settings. For measurement of percentages of neuroblast membrane with glia contact, we used a Fiji plugin as described previously [55]. Brains were

mounted with their dorsal side closest to coverslip, and all non-MB neuroblasts within 30 microns of the coverslip were assayed.

## Whole-brain Dilp-2 and Dilp-6 measurements

The CNS was dissected at 72 hours after larval hatching and lysed by sonication in extraction buffer (1× PBS with protease inhibitor). Lysed samples were cleared by centrifugation and dual-epitope tagged Dilp2 (Dilp2-HF) or Dilp6 (Dilp6-HF) was measured by ELISA [56]. Protein concentration for normalization was measured using a bicinchoninic assay (BCA) (Pierce, Thermo Fisher Scientific). To convert molar concentrations to pg Dilp/mg protein, we used a molecular weight of 7,828.86 for mature Dilp2$^{HF}$ protein (A-chain and B-chain with HA and FLAG tags) and 10,303.62 for mature Dilp6$^{HF}$ protein (A-chain, C-peptide, and B-chain with HA and FLAG tags).

## Quantitative RT-PCR

Total RNA was extracted from the 24-hour–fed larval CNS. For each experimental sample, 40 brains were pooled in TRIzol, and isolated RNA was reverse transcribed into cDNA using SuperScript IV First-Strand Synthesis System kit (Thermo Fisher Scientific). Quantitative PCR was performed using the iQ SYBR Green Supermix system (Bio-Rad, Hercules, CA, USA). *Gapdh1* expression was used as a control. Relative *dilp2* and *dilp6* mRNA levels for each sample were calculated using the 2-ΔΔCt method after normalizing to *Gapdh1* expression. The following primers were used: *dilp6* [45], *Gapdh1* [22], *dilp2* forward: ACGAGGTGCTGAGTATGGTGTGCG, and *dilp2* reverse: CACTTCGCAGCGG TTCCGATATCG.

## Statistical analysis

Student *t* tests and one-way ANOVAs were performed using Prism 8. For box plots, the boundary of the box closest to zero indicates the 25th percentile, a line within the box marks the median, and the boundary of the box farthest from zero indicates the 75th percentile. Whiskers (error bars) above and below the box indicate the maximum and minimum, respectively. The data in plots and the text are presented as means ± SEM.

## Supporting information

**S1 Fig. Glia and trachea endoreplicate in response to animal feeding.** (A) Larval brain cartoon with red box indicating the brain region imaged in Fig 1 and all subsequent Figs. (B–E) Single confocal Z images of non-MB and MB NBs. Top and bottom left panels are single-channel grayscale images, with colored overlay in the bottom right. Molecular markers are denoted within panels and white brackets indicate NBs in this and all figures. (F) Box and whisker plots of NB diameter of EdU-negative (black) and EdU-positive (orange) NBs. The number of NBs analyzed is indicated below plots. Student two-tailed *t* test, ***$p < 0.001$. (G) Fold changes in brain volume per hemisphere in response to animal feeding (see Materials and Methods). Column numbers indicate number of brain hemispheres scored. Values are normalized to 0-hour–fed animals. (H) Single Z-plane of a brain hemisphere. Left and middle panels are color overlays of a control brain after feeding. High-magnification images of 2 Repo-positive glia are shown to the right, one EdU-positive and one EdU-negative (white boxes). (I) Quantification of EdU-positive glial cells per brain lobe in response to feeding. (J) Number of Repo-positive glial cells per brain hemisphere before and after animal feeding. (K) Single Z image of a brain hemisphere from a *dupRNAi* knockdown animal. Left and middle panels are color

overlays with high magnification of 2 Repo-positive glia shown to the right (white boxes). (L) Quantification of glia number in *dupRNAi* knockdown animals compared to control. (M) Single Z image of a brain hemisphere. Left and middle panels are color overlays of control brain after feeding. A high-magnification image of a tracheal nucleus is shown to the right (white boxes). (N) Quantification of EdU in trachea from animals fed standard food or sucrose. (G,I) One-way ANOVA with Tukey post hoc analysis and (F,J,L,N) Student two-tailed *t* test, *$p < 0.05$, **$p < 0.01$, ***$p < 0.001$. Genotypes of panels listed in S2 Table and data listed in S1 Data. *dupRNAi*, *double-parked* RNA interference; EdU, 5-ethynyl-2′-deoxyuridine; MB, mushroom body; NB, neuroblast; Repo, reversed polarity.
(TIF)

**S2 Fig. PI3-kinase–dependent growth regulation in the developing brain.** (A) Maximum intensity projection of a single brain hemisphere labeled with EdU and quantification of NB size from indicated genotypes (B). (C,M) Glial and (E,H,I,K) tracheal morphology with quantification of glial surface area (D,N) and tracheal surface area from indicated genotypes (F,J,L). Top panels are colored overlays, with bottom panels showing single-channel grayscale images (A), single channel and mask (C,M), or rendered maximum intensity projection with segmentation of trachea (E,H,I,K). Molecular markers are denoted within panels. (G) Quantification of glial number. (B,D,F,J,L,N) Student two-tailed *t* test, *$p < 0.05$, **$p < 0.01$,***$p < 0.001$, error bars, SEM. (G) One-way ANOVA with Tukey post hoc analysis. Genotypes of panels listed in S2 Table and data listed in S1 Data. EdU, 5-ethynyl-2′-deoxyuridine; NB, neuroblast; PI3-kinase, phosphoinositide 3-kinase.
(TIF)

**S3 Fig. Glial identify based on location and GAL4 expression.** (A) Single Z image of a segmented brain hemisphere with glia subtypes and corresponding GAL4 lines. (B) Total number of glia subtypes before and after animal feeding. Glia type was identified based on location. Error bars, SEM. Black circles indicate single brain hemispheres. (C and E) Left panels, colored overlays of a single Z-plane from a brain hemisphere of the indicated genotype, with grayscale images on the right. Molecular markers are denoted within panels. Quantification of glial populations are shown in D and F. Red columns depict mean of histoneRFP-expressing glia, and black columns depict mean of glia identified based on position. Whites boxes indicate RFP-expressing cortex glia in (C) and SPG in (E). Genotypes of panels listed in S2 Table and data listed in S1 Data. RFP, red fluorescent protein; SPG, subperineurial glia.
(TIF)

**S4 Fig. Delayed NB reactivation and cortex glial growth in *dilp2* mutants is not due to delays in developmental timing.** (A) Box plots of NB diameter of EdU-negative (black) versus EdU-positive (orange) NBs. Column numbers indicate number of NBs analyzed. (B,C) Quantification of EdU-positive neuroblasts (B) and cortex glial membrane surface (C) area after 48 hours of feeding of indicated genotypes. (D,E) Mouth hook morphology of control and *dilp2* mutants at indicated times. (A–C) Student two-tailed *t* test, **$p < 0.01$. Genotypes of panels listed in S2 Table and data listed in S1 Data. Dilp, *Drosophila* insulin-like peptide; EdU, 5-ethynyl-2′-deoxyuridine; NB, neuroblast.
(TIF)

**S5 Fig. NB EdU incorporation in *dilp2*, *dilp6* double mutants.** (A,B) EdU-positive NBs after 24 hours of feeding in *dilp2*, *dilp6* double mutants. (A) Maximum intensity projection of a brain hemisphere. Top panel is a colored overlay (red, Dpn; green, EdU), and the bottom panel is a single-channel grayscale image (EdU) below with quantification in (B). Brain hemispheres are outlined, and the dotted vertical line indicates the midline. (B) One-way ANOVA

with Tukey post hoc analysis. ***$p < 0.001$. Genotypes of panels listed in S2 Table and data listed in S1 Data. Dilp, *Drosophila* insulin-like peptide; Dpn, Deadpan; EdU, 5-ethynyl-2′-deoxyuridine; NB, neuroblast.
(TIF)

**S1 Table. Classification of glial GAL4 expression in the larval brain after 24 hours of feeding.**
(DOCX)

**S2 Table. Genotypes listed by figure panel.**
(DOCX)

**S3 Table. *Drosophila* stock list and source.**
(DOCX)

**S1 Data. Excel spreadsheet with data listed for all main and supplementary figures.**
(XLSX)

## Acknowledgments

We thank Andrea Brand, Marc Freeman and the Bloomington and Kyoto stock centers for fly stocks and reagents. We especially thank Susan Doyle, Chhavi Sood, and Nahid Ausrafuggaman for help in tissue dissections. We thank Chris Doe, Susan Doyle, and Karsten Siller for providing comments on the manuscript.

## Author Contributions

**Conceptualization:** Xin Yuan, Michelle L. Bland, Sarah E. Siegrist.

**Data curation:** Xin Yuan, Sarah E. Siegrist.

**Formal analysis:** Xin Yuan, Conor W. Sipe, Sarah E. Siegrist.

**Funding acquisition:** Sarah E. Siegrist.

**Investigation:** Xin Yuan, Conor W. Sipe, Miyuki Suzawa, Michelle L. Bland, Sarah E. Siegrist.

**Methodology:** Xin Yuan, Conor W. Sipe, Miyuki Suzawa, Michelle L. Bland, Sarah E. Siegrist.

**Project administration:** Sarah E. Siegrist.

**Resources:** Sarah E. Siegrist.

**Software:** Sarah E. Siegrist.

**Supervision:** Michelle L. Bland, Sarah E. Siegrist.

**Validation:** Xin Yuan, Sarah E. Siegrist.

**Visualization:** Xin Yuan, Sarah E. Siegrist.

**Writing – original draft:** Xin Yuan, Sarah E. Siegrist.

**Writing – review & editing:** Xin Yuan, Conor W. Sipe, Michelle L. Bland, Sarah E. Siegrist.

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
