## [Editor Report · Decision Letter 0]

2 Oct 2019

Dear Dr Siegrist, 

Thank you for submitting your manuscript entitled "Dilp-2 mediated PI3-kinase activation coordinates reactivation of quiescent neuroblasts with growth of their glial stem cell niche" for consideration as a Research Article by PLOS Biology.

Your manuscript has now been evaluated by the PLOS Biology editorial staff [as well as by an academic editor with relevant expertise] and I am writing to let you know that we would like to send your submission out for external peer review.

Please re-submit your manuscript within two working days, i.e. by Oct 04 2019 11:59PM.

Kind regards,

Di Jiang,

Associate Editor

PLOS Biology

---

## [Decision Letter · Decision Letter 1]

5 Nov 2019

Dear Dr Siegrist,

Thank you very much for submitting your manuscript "Dilp-2 mediated PI3-kinase activation coordinates reactivation of quiescent neuroblasts with growth of their glial stem cell niche" for consideration as a Research Article at PLOS Biology. Your manuscript has been evaluated by the PLOS Biology editors, an Academic Editor with relevant expertise, and by four independent reviewers.

In light of the reviews (below), we will not be able to accept the current version of the manuscript, but we would welcome resubmission of a much-revised version that takes into account the reviewers' comments. You will need to re-write the paper including its Abstract, Introduction, Results, and Discussion to inform the reader clearly what are the novel results and what aren't. You should emphasise the novel results and expand them while moving the confirmation experiments to supplementary information to ensure the distinction. We will want to see all experiments suggested by reviewer 3. You will also need to address several points raised by reviewer 4 by more careful analyses of the experiments you have already done and more discussion. You should also respond to this reviewer's points related to the sucrose effect through better introduction and contextualisation of previous studies on neuroblast reactivation in response to dietary amino acids and/or additional experiments. We cannot make any decision about publication until we have seen the revised manuscript and your response to the reviewers' comments. Your revised manuscript is also likely to be sent for further evaluation by the reviewers.

Your revisions should address the specific points made by each reviewer. Please submit a file detailing your responses to the editorial requests and a point-by-point response to all of the reviewers' comments that indicates the changes you have made to the manuscript. In addition to a clean copy of the manuscript, please upload a 'track-changes' version of your manuscript that specifies the edits made. This should be uploaded as a "Related" file type. You should also cite any additional relevant literature that has been published since the original submission and mention any additional citations in your response. 

Before you revise your manuscript, please review the following PLOS policy and formatting requirements checklist PDF: http://journals.plos.org/plosbiology/s/file?id=9411/plos-biology-formatting-checklist.pdf. It is helpful if you format your revision according to our requirements - should your paper subsequently be accepted, this will save time at the acceptance stage.

Please note that as a condition of publication PLOS' data policy (http://journals.plos.org/plosbiology/s/data-availability) requires that you make available all data used to draw the conclusions arrived at in your manuscript. If you have not already done so, you must include any data used in your manuscript either in appropriate repositories, within the body of the manuscript, or as supporting information (N.B. this includes any numerical values that were used to generate graphs, histograms etc.). For an example see here: http://www.plosbiology.org/article/info%3Adoi%2F10.1371%2Fjournal.pbio.1001908#s5.

For manuscripts submitted on or after 1st July 2019, we require the original, uncropped and minimally adjusted images supporting all blot and gel results reported in an article's figures or Supporting Information files. We will require these files before a manuscript can be accepted so please prepare them now, if you have not already uploaded them. Please carefully read our guidelines for how to prepare and upload this data: https://journals.plos.org/plosbiology/s/figures#loc-blot-and-gel-reporting-requirements.

Upon resubmission, the editors will assess your revision and if the editors and Academic Editor feel that the revised manuscript remains appropriate for the journal, we will send the manuscript for re-review. We aim to consult the same Academic Editor and reviewers for revised manuscripts but may consult others if needed.

We expect to receive your revised manuscript within two months. Please email us (plosbiology@plos.org) to discuss this if you have any questions or concerns, or would like to request an extension. At this stage, your manuscript remains formally under active consideration at our journal; please notify us by email if you do not wish to submit a revision and instead wish to pursue publication elsewhere, so that we may end consideration of the manuscript at PLOS Biology.

When you are ready to submit a revised version of your manuscript, please go to https://www.editorialmanager.com/pbiology/ and log in as an Author. Click the link labelled 'Submissions Needing Revision' where you will find your submission record. 

Sincerely,

Di Jiang

PLOS Biology

Reviewer remarks:

Reviewer #1: Review of “Dilp-2 mediated PI3-kinase activation coordinates reactivation of quiescent neuroblasts with growth of their glial stem cell niche”

Author summary:

Here, we report that Drosophila neural stem cells, known as neuroblasts, reactivate from developmental quiescence in a dietary nutrient-dependent manner. Neuroblast reactivation requires non-cell autonomous activation of PI3-kinase signaling from cortex glia and tracheal processes, both of which are closely associated with neuroblasts. Furthermore, PI3-kinase activation in neuroblasts is required non-cell autonomously for glial membrane expansion and robust neuroblast-glial contact. Finally, PI3-kinase is required cell autonomously for nutrient dependent growth of neuroblasts, glia, and trachea. Of the seven Drosophila insulin-like peptides (Dilps), we find that Dilp-2 is required for PI3-kinase activation and growth coordination between neuroblasts and glia. Dilp-2 induces cortex glia to initiate membrane growth and make first contact with quiescent neuroblasts. After contact, neuroblasts increase in size and reenter S-phase. Once reactivated from quiescence, neuroblasts promote growth of cortex glia which in turn form a selective membrane barrier around neuroblasts and their newborn progeny.

I really liked this paper. The results are novel, particularly the role of trachea in signaling to neuroblasts. The results are well-supported by the figures and statistics provided. The figures are clear and highlight the relevant phenotypes in a convincing manner.

Major comments

None

Minor comments

1. Consider splitting the quite long results section “Nutrient-dependent growth of neuroblasts, glia, and trachea requires cell autonomous and cell non-autonomous activation of PI3-kinase” into two sections, one for cell autonomous data and one for cell non-autonomous data. This is optional at author discretion. 

2. If the authors have data on knockdown of PI3K in both glia and trachea they should include it. These two cell types are likely functioning semi-redundantly in signaling to neuroblasts.

3. Several glial subtype Gal4 lines characterized on page 8; why was the well-known astrocyte-specific alrm-Gal4 line excluded? Just curious. 

4. Page 8 second line from bottom. I think “compared” should be “comparable”

5. Page 10 second paragraph says “growth coordination between cortex glia and neuroblasts is Dilp-2 dependent” – which raises the question of what is the effect of creating a mismatch in growth coordination where neuroblasts and glia grow differently? Is there an effect on viability? Brain development?

Reviewer #2: Xin Yuan et al. described a role of Dilp-2 mediated PI3-kinase activation in coordinating reactivation of quiescent neuroblasts with growth of glial niche. They reported that nutrient-dependent growth of neuroblasts, glia and trachea requires cell autonomous and cell non-autonomous activation of PI3-kinase. They went on to show that several different subtypes of glia, including cortex glia, subperineural glia (SPG) and neuropil glia, require dietary nutrients and PI3-kinase activity for growth. The authors found that cortex glia are required to reactivate neuroblasts and that Dilp-2 regulates neuroblast reactivation, cortex glia membrane growth and Dilp-6 protein levels. Finally, they demonstrated that increases in neuroblast membrane with glia contact precede increases in neuroblast size, implying that cortex glia membrane contact triggers neuroblast growth and reactivation. Overall, they presented an impressive amount of data, some of which are very interesting.

Major concern

This manuscript significantly overlapped with data previously reported (Pauline Speder and Andrea Brand, Elife, January 2018) in the following conclusions. 1) Neuroblast reactivation induces remodelling of cortex glia; 2) Nutrition-induced activation of PI3K/Akt drives the cortex glia to expand their membrane processes; 3) Dilps are required for development of cortex glia. One major claim of the current manuscript is the role of cortex glia in neuroblast reactivation, which contradicts with the previous conclusion that cortex glia has little impact on neuroblast reactivation (Elife 2018). The relevant findings on development/remodelling of cortex glia associated with neuroblast reactivation reported on Elife 2018 was wrongly referred to as SPG glia in the introduction on page 3 of this manuscript. The authors are advised to remove duplicated data and to provide the reasons underlying these different conclusions on the role of cortex glia in neuroblast reactivation.

Reviewer #3: In this paper Yuan and coworkers address how different cell types coordinate their growth in response to dietary nutrients. They use neuroblasts in the fly brain since these neural stem cells reactivate from quiescence in a nutrient dependent manner and are in close contact with other tissues in the niche (i.e. glia and trachea). The main claims of the paper are:

1) that the coordinated growth between neuroblast and cortex glia is based on a bidirectional growth signaling between these two tissues. 

2) that dilp2 is required for neuroblast reactivation and cortex glia growth and hence the coordination of the two processes. 

The idea in first claim is not completely novel. Speder and Brand. eLife 2018;7:e30413 have shown that cortex glia growth in the niche is nutrient dependent and that it requires autonomous activation of PI3K . They also acknowledge the existence of another signal regulating cortex glia growth and link it to neuroblast reactivation. Yuan et al. main contribution to claim 1 is to propose that this signal is Dilp2 dependent PI3K activation in the neuroblast. If besides this contribution, the authors consider that their results add more to this claim, this was not evident in the text and they might want to emphasize any points that add to Speder and Brand findings. 

The second claim is novel and is actually reflected in the title of the manuscript. Sousa-Nunes et al. 2011, Chell and Brand 2010, have shown that Dilps are involved in neuroblast reactivation (dilp2, 3, 5, 6) with some being expressed in glia (dilp3, 6), this paper addresses which dilp is required to coordinate the reactivation of neuroblasts and the growth of cortex glia. Through loss of function analyses they propose that dilp2 mediates neuroblast reactivation and cortex glia growth. They carry out a detailed analysis showing that an increase in glia-neuroblast membrane contacts increase takes place prior to neuroblast reactivation. 

These are interesting findings that would gain strength and benefit from additional experiments. 

Revision of the following major points is essential prior to publication: 

1. The loss of function analysis of Dilps is done with null alleles generated though homologous recombination. It seems, based on the genotype provided, that they are analyzed in homozygous conditions. This has the risk of carrying additional 2nd site mutations in homozygosis. It would be reassuring to verify the Dilp2 results for neuroblast EdU incorporation and cortex glia growth with an additional allelic combination removing Dilp2 using two independent mutant chromosomes (dilp2/def or two dilp2 mutants from independent homologous recombination events). 

2. Dilp2 null mutant has been reported to have a developmental delay. A time line experiment analyzing reduction in EdU incorporation and cortex glia growth at later time points than 24hrs of feeding could distinguish whether the phenotypes observed are sustained in time or a consequence of the developmental delay. 

3. Cell type specific rescue experiments could identify the source of Dilp2, test sufficiency and specificity (i.e. would dilp6 overexpression rescue the dilp2 phenotype?). 

While Sousa-Nunes et al. 2011, Chell and Brand 2010 are referenced in the manuscript, a better description of what is already known about Dilp involvement in neuroblast reactivation might help this manuscript to highlight their own contributions, and make it more accessible to non-specialists. The details of the methodology are sufficient to allow the experiments to be reproduced. 

Minor points: 

1. In the second block of results “Nutrient dependent growth …”

- When levels of PI3-kinase activity were reduced in neuroblasts (worGAL4, UAS dp60), neuroblast EdU incorporation was reduced and neuroblast size reduced as previously reported (Figure 2B and S2A) → It would help if the reader were referred to compare S2A to Figure 1J for the Edu incorporation reduction

- When levels of PI3-kinase activity were reduced in all glia (repoGAL4, UAS-dp60), EdU incorporation was essentially absent in glia and membrane surface area reduced (Figure 2C and S2B,C). → It would help if the reader were referred to compare S2B to Figure 1P for glia membrane surface area reduction. It is unclear how the information that S2C provides fits here 

- Reduction of PI3-kinase activity in trachea (btlGAL4, UAS-dp60), eliminated EdU incorporation and tracheal surface area and branching were reduced (Figure 2D and S2D). → It would help if the reader were referred to compare S2D to Figure 1V for tracheal phenotypes. 

EdU experiments mentioned above are not shown for either glia or trachea, the way it is written makes the reader expect the data in the referred figures. It would be desirable to see these data in the supplementary figure. 

- Figure 2N, O: The Figure panels in do not match the figure legend.

2. Discussion 

- PVF was meant to mean PVR? Please spell out acronym. 

3. Figure legend 6

- (H) One-way ANOVA with Tukey’s post hoc analysis and (K) Student’s two-tailed t test → Is K supposed to be L?

4. Figure 7

- glia (repoGAL4,UAS-mCD8:GFP) at the top of panel A → does not match the mCD8:RFP labeling indicated in panel A or the genotype of these animals as specified in S5 table.

Reviewer #4: The authors use a series of cell specific alterations of the PI3K pathway to define both autonomous and non-autonomous relationships between neuroblasts, glia, and trachea in regulating their growth. The paper is well written and the experiments are elegant. Overall, the findings of the manuscript provide new and detailed insights into the crosstalk between cells in the brain that are required for proper growth of the brain in response to nutrition. My specific comments can be found below.

To determine whether the gial have undergone endo-reduplication the authors could DAPI quantify their nuclei.

Do trachea undergo endo-reduplication?

It would be helpful at the end of the results section relevant to Figure 1 to summarize the specific changes that occur in response to feeding and their relative timing. 

How does NB diameter in sucrose fed animals, compare to worGal4 UAS-dp60? 

How does glial membrane surface area in sucrose fed animals, compare to repoGal4 UAS-dp60?

How does tracheal surface area in sucrose fed animals, compare to btlGal4 UAS-dp60?

Why do the authors think that glial membrane growth reduction is only modest after inactivation of the PI3 kinase pathway in NBs? Do they think there are other signals or do they think the expression of dp60 is not strong enough to inactivate the PI3 Kinase pathway?

Why do the authors think that NB incorporation of EdU is only modest after inactivation of the PI3 kinase pathway in trachea? Do they think there are other signals or do they think the expression of dp60 is not strong enough to inactivate the PI3 Kinase pathway?

What happens to NB size after inactivation of the PI3 kinase pathway in trachea?

A table that summarizes the outcome of the experiments described in Figure 2 would be helpful.

“Somewhat surprisingly, neuroblast EdU incorporation in dilp6 null mutants was not changed ...”

Why is this surprising?

Why is there an affect on NB incorporation of EdU in dilp1 and in dilp7 mutants? In other words, where would the authors put dilp1 and dilp7 relative to dilp2 mechanistically or at least genetically?

Can cell specific activation of the PI3 kinase pathway in one cell type rescue non-autonomously phenotypes seen in other cell types in animals mutant for dilps, or fed sucrose? 

“Trachea allow for oxygen exchange, but are also important sources of cell signaling molecules, including EGF-R and FGF-R.”

What are EGF-R and FGF-R? Do they mean the ligands or the receptors?

---

## [Decision Letter · Decision Letter 2]

1 Apr 2020

Dear Dr Siegrist,

Thank you for submitting your revised Research Article entitled "Dilp-2 mediated PI3-kinase activation coordinates reactivation of quiescent neuroblasts with growth of their glial stem cell niche" for publication in PLOS Biology. I have now obtained advice from three of the four original reviewers and have discussed their comments with the Academic Editor. 

Based on the evaluations, we're delighted to let you know that we're now editorially satisfied with your manuscript. However before we can formally accept your paper and consider it "in press", we also need to ensure that your article conforms to our guidelines. A member of our team will be in touch shortly with a set of requests. As we can't proceed until these requirements are met, your swift response will help prevent delays to publication. Please also make sure to address the data and other policy-related requests noted at the end of this email.

*Copyediting*

*Published Peer Review History*

*Early Version*

*Submitting Your Revision*

Sincerely,

Di Jiang, PhD 

Associate Editor

PLOS Biology

DATA POLICY:

Regardless of the method selected, please ensure that you provide the individual numerical values that underlie the summary data displayed in the following figure panels as they are essential for readers to assess your analysis and to reproduce it: Figures 1FLR, 2EHIK, 3DH, 4DEHIL, 5AGJ, 6CF-K, 7CDK, S1FGIJLN, S2BDFGJLN, S3ADF. NOTE: the numerical data provided should include all replicates AND the way in which the plotted mean and errors were derived (it should not present only the mean/average values).

Reviewer remarks:

Reviewer #1: The authors have addressed my points satisfactorily. I liked the first version quite a bit, and it is even better noe. In addition, I appreciate the effort taken by reviewers 2-4, and I feel the authors have addressed the vast majority of their comments too, which makes this a much improved paper. I am strongly in favor of acceptance and rapid publication.

Reviewer #3: The authors have satisfactorily addressed all my concerns, both by rewriting sections of the manuscript and performing the experiments suggested. These changes have improved the manuscript by better highlighting and differentiating their work from previously published articles. 

Reviewer #4: The authors have adequately addressed my comments from the first review. I recommend publication.

---

## [Editor Report · Decision Letter 3]

13 May 2020

Dear Dr Siegrist,

On behalf of my colleagues and the Academic Editor, Catarina C Homem, I am pleased to inform you that we will be delighted to publish your Research Article in PLOS Biology. 

Early Version

PRESS 

Kind regards,

Alice Musson

Publishing Editor, 

PLOS Biology

on behalf of

Di Jiang,

Associate Editor

PLOS Biology